# Mutational robustness changes during long-term adaptation in laboratory budding yeast populations

Milo S Johnson[1,2,3]*, Michael M Desai[1,2,3,4]*

[1]Department of Organismic and Evolutionary Biology, Harvard University, Cambridge, United States; [2]Quantitative Biology Initiative, Harvard University, Cambridge, United States; [3]NSF-Simons Center for Mathematical and Statistical Analysis of Biology, Harvard University, Cambridge, United States; [4]Department of Physics, Harvard University, Cambridge, United States

**Abstract** As an adapting population traverses the fitness landscape, its local neighborhood (i.e., the collection of fitness effects of single-step mutations) can change shape because of interactions with mutations acquired during evolution. These changes to the distribution of fitness effects can affect both the rate of adaptation and the accumulation of deleterious mutations. However, while numerous models of fitness landscapes have been proposed in the literature, empirical data on how this distribution changes during evolution remains limited. In this study, we directly measure how the fitness landscape neighborhood changes during laboratory adaptation. Using a barcode-based mutagenesis system, we measure the fitness effects of 91 specific gene disruption mutations in genetic backgrounds spanning 8000–10,000 generations of evolution in two constant environments. We find that the mean of the distribution of fitness effects decreases in one environment, indicating a reduction in mutational robustness, but does not change in the other. We show that these distribution-level patterns result from differences in the relative frequency of certain patterns of epistasis at the level of individual mutations, including fitness-correlated and idiosyncratic epistasis.

*For correspondence:
milo.s.johnson.13@gmail.com
(MSJ);
mdesai@oeb.harvard.edu (MMD)

**Competing interest:** The authors declare that no competing interests exist.

## Editor's evaluation

Johnson and Desai developed an innovative yeast experimental-evolution system where they can insert barcoded disruptive mutations into the genome and measure their individual effect on fitness. They use this system to test whether these mutations have different effects on evolving lineages as they adapt over time. As expected, the mean fitness effect does decline in most (but not all) populations as lineages adapt, but in another condition, mean fitness effects of mutations do not change as the populations adapt. The authors suggest an intriguing interpretation that the 'control coefficient' of selection on growth can shift between different genetic modules over time, resulting in differing magnitudes of epistasis.

## Introduction

Evolutionary adaptation relies on recombination and spontaneous mutagenesis to constantly introduce variation into populations, upon which natural selection can act. The fate of a single mutation – and its impacts on the dynamics of adaptation – depends on how it affects organismal fitness, which we know depends in complex ways on the rest of the genetic background (reviewed in *de Visser et al., 2011*; *Domingo et al., 2019*; *Lehner, 2011*). Understanding this genotype dependence, or

epistasis, involves analyzing key features of the fitness landscape, the high-dimensional map between genotype and fitness (*Wright, 1932*).

To investigate these questions, numerous studies have surveyed epistatic interactions among a variety of different types of mutations. These studies have found that beneficial mutations isolated from laboratory evolution experiments tend to exhibit negative epistasis. That is, they are less beneficial in combination than would be expected from the combination of their effects in isolation (*Karkare et al., 2021*; *MacLean et al., 2010*; *Ono et al., 2017*; *Rokyta et al., 2011*; *Schenk et al., 2013*; *Schoustra et al., 2016*; *Zee and Velicer, 2017*). However, some examples of positive epistasis among beneficial mutations have also been observed (*Chou et al., 2009*; *Fumasoni and Murray, 2020*; *Hsieh et al., 2020*; *Khan et al., 2011*; *Levin-Reisman et al., 2019*). Surveys of interactions between deleterious mutations have produced numerous examples of both positive and negative epistasis (*Costanzo et al., 2016*; *Elena and Lenski, 1997*; *Hall and MacLean, 2011*; *Jasnos and Korona, 2007*; *Lalić and Elena, 2012*; *Sanjuán et al., 2004*; *Van Leuven et al., 2021*).

While this earlier work has identified a wide range of epistatic interactions among specific combinations of mutations, several recent studies of epistasis in laboratory microbial systems have found that general patterns of *fitness-correlated epistasis* often emerge. These fitness-correlated patterns typically favor less-fit genotypes: for both beneficial and deleterious mutations, we usually find that the fitness effect of a mutation is negatively (rather than positively) correlated with the fitness of the genetic background on which it occurs (*Chou et al., 2011*; *Johnson et al., 2019*; *Khan et al., 2011*; *Kryazhimskiy et al., 2014*). These patterns have been termed *diminishing returns* and *increasing costs* for beneficial and deleterious mutations, respectively. These trends are particularly intriguing to those interested in modeling the dynamics of adaptation for two reasons. The first is the analytical promise of a simple, monotonic relationship between fitness and the fitness effects of mutations. The second is the potential for these patterns of epistasis to explain declining adaptability, a commonly observed phenomenon in laboratory evolution experiments in which the rate of fitness increase slows as evolution proceeds (reviewed in *Couce and Tenaillon, 2015*; see *Wünsche et al., 2017* for an analysis of the link between diminishing returns and declining adaptability).

The initial studies that identified patterns of fitness-mediated epistasis did so in the context of a relatively small number of beneficial mutations, finding that these mutations became systematically less beneficial in more-fit genetic backgrounds. More recent work has shown that as populations evolve the average fitness effect of a spontaneous beneficial mutation decreases (*Aggeli et al., 2021*; *Wünsche et al., 2017*). Much less work has been done to characterize how the effects of deleterious mutations interact with the beneficial mutations that fix during evolution (*Remold and Lenski, 2004*). We recently showed that the fitness effects of larger panels of ~100–1000 random insertion mutations (both beneficial and deleterious) tend to be systematically less beneficial or more deleterious in more-fit backgrounds (*Johnson et al., 2019*). However, the genetic backgrounds in that study were derived from a cross between two diverged yeast strains. It remains unclear whether similar patterns would hold across genetic backgrounds that are the result of long-term laboratory evolution because the mutations that drive evolutionary adaptation are selected along a line of descent, which in principle could affect their epistatic interactions.

Here, we directly address this question by measuring the fitness effects of a panel of insertion mutations during the course of a long-term laboratory evolution experiment in budding yeast. Specifically, we isolated clones from six timepoints spanning 8000–10,000 generations of adaptation to each of two constant laboratory environments from the ongoing evolution experiment we have recently described (*Johnson et al., 2021*). While the yeast strains used in our prior experiment differed at tens of thousands of segregating loci, the strains in this experiment differ by only tens or hundreds of mutations that fixed successively during evolution. By looking at the effects of insertion mutations in these strains, we are measuring a panel of hidden phenotypes (the fitness effects of the mutations) that may change predictably or stochastically during evolution. The widespread presence of epistasis observed in biological systems suggests that these hidden phenotypes may be important to long-term evolutionary outcomes as the fitness effects of mutations effectively open and close doors to unique pathways for evolution (*Johnson et al., 2021*; *Karkare et al., 2021*; *Kvitek and Sherlock, 2011*).

Robustness can be broadly defined as invariance in the face of perturbation (*Masel and Siegal, 2009*). Here, we are concerned specifically with mutational robustness, a measure of how invariant phenotypes are to mutations (*Lauring et al., 2013*). Our approach complements several recent

studies that have analyzed changes in mutational robustness during evolution by conducting muta-genesis followed by phenotypic measurements. For example, *Novella et al., 2013* found that vesicular stomatitis virus strains evolved in the lab gained robustness, measured based on survival after mutagenesis. In contrast, *Butković et al., 2020* found that a strain of turnip mosaic potyvirus evolved in *Arabidopsis thaliana* lost robustness over time, measured as the change in the ability of the virus to retain its level of plant infectivity after mutagenesis. Our barcode-based mutagenesis system makes it possible to dissect these overall changes in robustness by measuring how they emerge as a result of changes in the effects of individual mutations.

In this study, we aim to leverage this mutation-level data to better understand the structure of epistasis in evolving populations. First, we analyze the overall distribution of fitness effects to measure how mutational robustness changes during adaptation in each environment. Next, we ask whether the distribution-level changes we observe can be explained by patterns of fitness-correlated epistasis among individual mutations. Finally, we examine how the effects of these mutations change in each of the evolving populations, asking whether epistasis is more often driven by predictable adaptations common across populations or by specific mutations fixed in a single population.

## Results
### Changes in the distribution of fitness effects during evolution
We isolated two clones from each of six timepoints from six haploid populations evolved in rich media at a permissive temperature (YPD 30°C) and from six haploid populations evolved in a defined media environment at a high temperature (SC 37°C), a total of 144 clones. We measured the fitness of each of these clones in the environment to which they adapted, finding that fitness increases steadily through time in both the YPD 30°C and SC 37°C environments, and displays a general pattern of declining adaptability (*Figure 1A*).

We next created a library containing each of 91 insertion mutations in each of our 144 clones. This set of mutations was identified previously as a subset of random insertion mutations that have measurable effects in the strains from *Johnson et al., 2019*. These mutations have a similar spectrum of effects in the clones isolated for this study (*Figure 1—figure supplement 8*), suggesting that they are also a broad sample of insertion mutations with measurable fitness effects in these strains. We measured the fitness effect of each mutation in each clone using barcode-based competition assays as described in *Johnson et al., 2019*. Because the molecular dynamics of evolution in these haploid populations are characterized by successive selective sweeps, we expect the two clones isolated from each population at each timepoint to have very similar genetic backgrounds. When we compare the average fitness effect measurement for each insertion mutation between these clones, we generally see strong agreement, with a few exceptions (*Figure 1—figure supplements 1–3*). These exceptions likely represent rare but important genetic differences between clones from the same population-timepoint. Given this, we chose to analyze our data in two ways. First, we improve the reliability of our fitness effect measurements for each population-timepoint by using measurements from combined barcodes (cBCs) from both clones, treating them as we treated biological replicates in *Johnson et al., 2019*. Second, we treat each clone independently. *Figure 1—figure supplement 7*, *Figure 2—figure supplement 5*, *Figure 3—figure supplement 5*, and *Figure 4—figure supplement 1* show that our qualitative conclusions are unchanged when using this second approach.

We find that the distribution of fitness effects (DFE) of our 91 insertion mutations changes over the course of the evolution experiment. In the YPD 30°C environment, the mean of the DFE decreases over time as fitness increases during evolution (*Figure 1B*), consistent with a general pattern of both diminishing returns and increasing costs. The negative relationship between generations evolved and the mean of the DFE is significant in the entire set of population-timepoints ($p=2.0 \times 10^{-6}$, Wald test) and in four of our six individual populations ($p<0.05$, Wald test). In contrast, although the DFE does shift in individual populations evolved in our SC 37°C environment, we do not see any consistent patterns (*Figure 1B*).

In both environments, we find that the changes in the mean of the DFE are modest compared to our previous experiment (*Figure 1—figure supplement 6* shows that the slope of the regression between the DFE mean and background fitness in YPD 30°C is less than half the magnitude of the corresponding slope in *Johnson et al., 2019*). The reasons for this are unclear, but may reflect the fact that

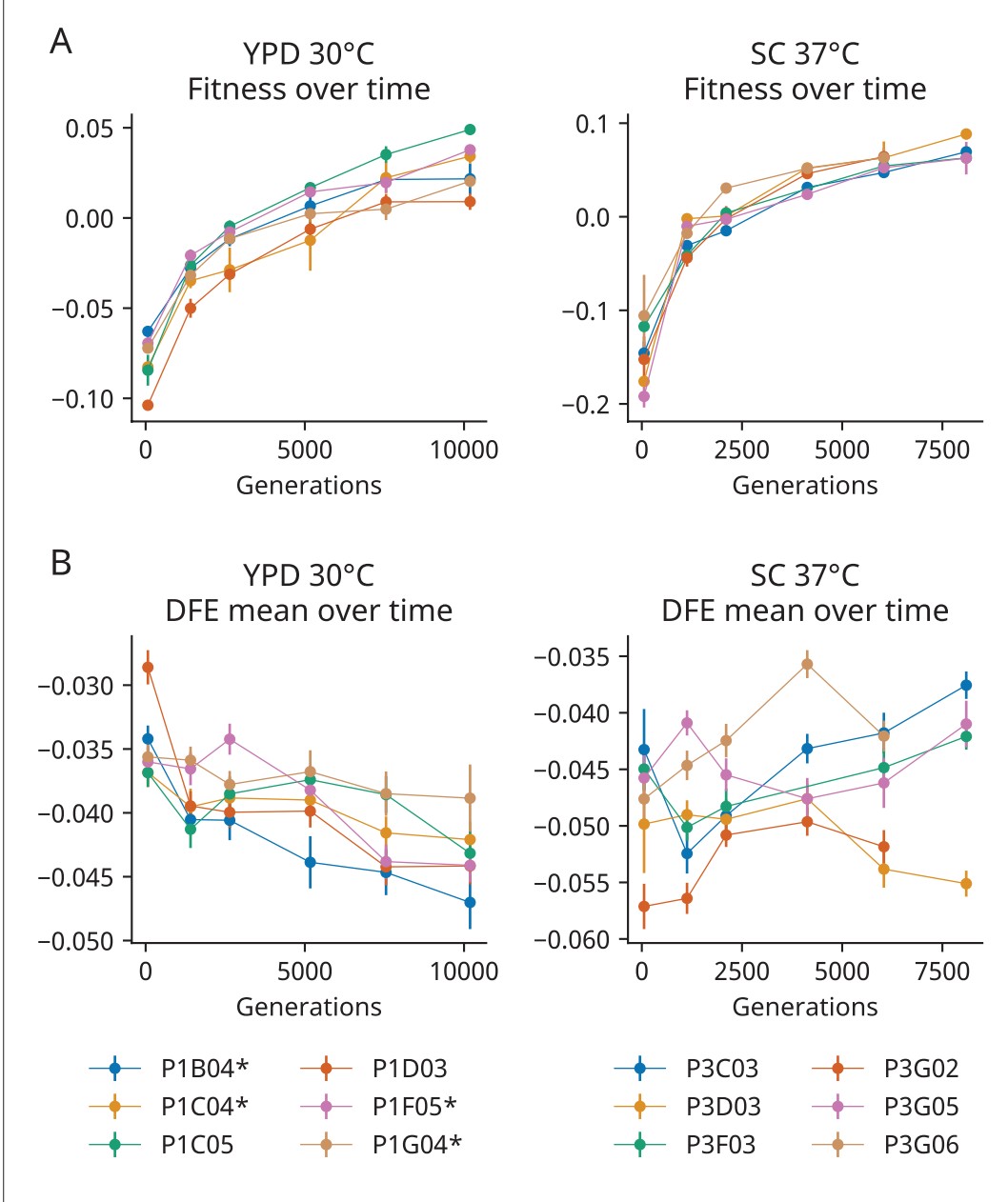

**Figure 1.** The distribution of fitness effects (DFE) mean declines in one of two environments during evolution. (**A**) Changes in fitness during the evolution experiment, measured as the average fitness of two clones isolated from six timepoints in each population. In each graph, zero is the fitness of a fluorescent reference used in that environment. Error bars represent the standard deviation of the fitnesses measured for the two clones (points without error bars have errors smaller than the point size). (**B**) The mean fitness effect of the insertion mutations measured in the clones isolated from each timepoint. Asterisks represent a significant correlation (p<0.05, Wald test) between generation and DFE mean in that population alone. In both (**A**) and (**B**), the six populations are indicated by color. Error bars represent the standard error of the DFE mean, calculated from the standard errors of individual mutations (see 'Materials and methods'). Additional DFE statistics are shown in *Figure 1—figure supplement 4*.

The online version of this article includes the following figure supplement(s) for figure 1:

**Figure supplement 1.** Fitness effect measurement correlations in YPD 30°C.

**Figure supplement 2.** Fitness effect measurement correlations in SC 37°C.

**Figure supplement 3.** Fitness effect measurement correlations in clones evolved in YPD 30°C, assayed in SC 37°C.

*Figure 1 continued on next page*

*Figure 1 continued*

**Figure supplement 4.** Additional distribution of fitness effects (DFE) statistics.

**Figure supplement 5.** Accounting for missing fitness effect measurements.

**Figure supplement 6.** Comparison of our distribution of fitness effects (DFE) mean vs. background fitness data with the data from *Johnson et al., 2019*.

**Figure supplement 7.** Distribution of fitness effects (DFE) statistics and missing fitness effect measurements for analysis considering clones separately.

**Figure supplement 8.** Distributions of all fitness effects measured in *Johnson et al., 2019* and this experiment.

**Figure supplement 9.** Excluding barcodes that experience sequencing cross-contamination.

fitness differences between the strains we study here are caused by a smaller number of mutations. We note that because of the modest differences in the DFE between clones, the noise in our measurements contributes significantly to the variation. One potentially biased source of this noise is missing measurements: not all mutations have fitness effects measured in each population-timepoint due to differences in transformation efficiency or, most commonly, mutations that had too few read counts in the first two timepoints of the fitness assay. As in *Johnson et al., 2019*, missing measurements of strongly deleterious mutations are more common in more-fit strains (clones from later timepoints) in YPD 30°C, suggesting that the negative correlation between generations evolved and the mean of the DFE would be stronger if more of these deleterious mutations had been successfully measured in clones from later timepoints. Indeed, if we look at a limited set of mutations with measurements in every population-timepoint or if we 'fill in' missing measurements using their average fitness effect across population-timepoints, we see similar or stronger patterns of change ($p<5 \times 10^{-7}$, Wald test) in the DFE mean in YPD 30°C (*Figure 1—figure supplement 5*). This relationship between the DFE mean and generations evolved also holds ($p<5 \times 10^{-8}$, Wald test) in our analysis where clones are treated independently (*Figure 1—figure supplement 7*).

## Epistasis at the level of individual mutations

We next turn to the components of these distribution-level patterns: epistatic patterns for individual mutations. To look at these patterns, we focus on mutations that have fitness measurements in at least 20 population-timepoints in each environment (77 mutations in YPD 30°C, 74 in SC 37°C, 70 in both). We start by looking for patterns of fitness-correlated epistasis. We classify each mutation in each environment as correlated negatively or positively with background fitness if the correlation is significant at the $p<0.05$ level (Wald test) and the absolute value of the slope is greater than 0.05, and classify them as not significantly correlated if they do not meet these criteria. The cutoff of 0.05 for the slope was chosen to filter for mutations with an effect size across the range of background fitness that is larger than the typical standard error for our fitness effect measurements (see 'Materials and methods'). For both environments, we find numerous examples of both negative and positive correlations, corresponding to increasing costs or decreasing costs, respectively, for deleterious mutations (and to diminishing returns or increasing returns, respectively for beneficial mutations).

In *Figure 2*, we show examples illustrating how the fitness effect of specific mutations vary across clones isolated from each environment, plotted as a function of the fitness of each clone in that environment (i.e., the background fitness). We find numerous examples of mutations that exhibit negative, positive, and nonsignificant correlations with background fitness in both environments (panels along the diagonal). We also find examples where a specific mutation exhibits a nonsignificant correlation in one environment and either a positive or negative correlation in the other (off-diagonal panels). Overall, consistent with our previous results, we observe about twice as many negative correlations as positive correlations. As we would expect from the DFE-level results, this imbalance is more pronounced in the YPD 30°C environment: 33/77 (~43%) mutations have fitness effects that decline significantly as background fitness increases in YPC 30°C compared to 17/74 (~23%) in SC 37°C. We also find that only 9/77 (~12%) mutations display the opposite pattern (fitness effects increase significantly as background fitness increases) in YPD 30°C, while 13/74 (~18%) display this pattern in SC 37°C. Because we are primarily focused on comparing the frequency of each pattern across environments, we report these values before multiple-hypothesis-testing correction here and in *Figure 2*;

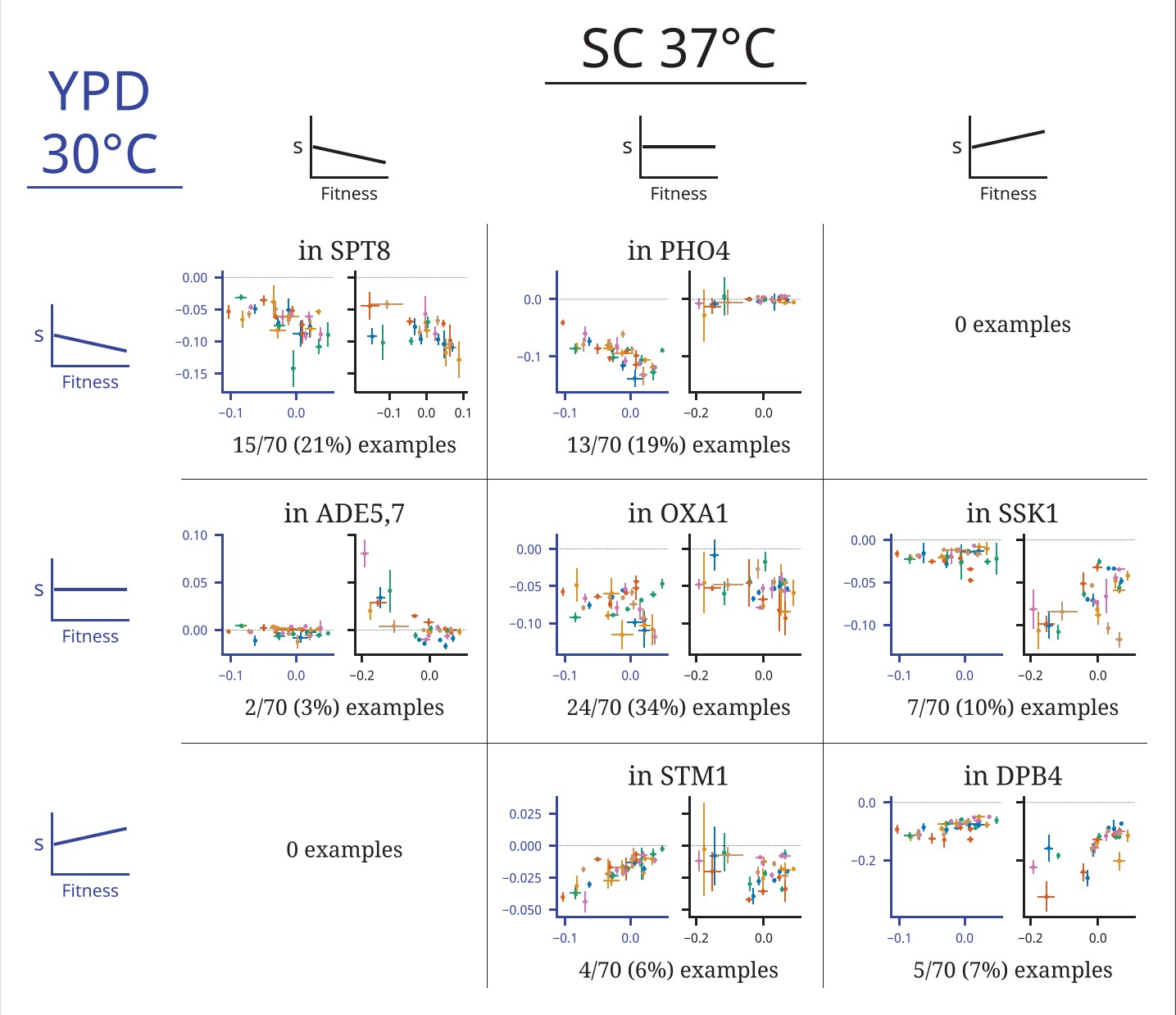

**Figure 2.** Patterns of fitness-correlated epistasis. Each panel shows an example of a specific mutation with a particular combination of relationships (negative, positive, or nonsignificant correlation between fitness effect of the mutation, $s$, and background fitness) in the two environments; numbers indicate the total number of mutations displaying each pair of relationships. Each point depicts the fitness effect (y-axis) of one insertion mutation measured in one population-timepoint, with the measured fitness of that population-timepoint represented on the x-axis. Error bars show the standard error of both measurements (see 'Materials and methods'). Axes are colored to identify the environment: in each panel, the blue axes on the left are data from YPD 30°C and the black axes on the right are data from SC 37°C. Points are colored by population, as in *Figure 1*. Each set of example plots is labeled by where the mutation is in the genome (i.e., which gene it disrupts). Additional comparisons of patterns of epistasis in these experiments and those from *Johnson et al., 2019* are shown in *Figure 2—figure supplements 1–4*.

The online version of this article includes the following figure supplement(s) for figure 2:

**Figure supplement 1.** Comparison of patterns of fitness-correlated epistasis between YPD 30°C and a previous study.

**Figure supplement 2.** Comparison of patterns of fitness-correlated epistasis between SC 37°C and a previous study.

**Figure supplement 3.** Comparison of patterns of fitness-correlated epistasis YPD 30°C and SC 37°C, in both cases using the set of clones isolated from evolution in YPD 30°C.

**Figure supplement 4.** Comparison of patterns of fitness-correlated epistasis in the SC 37°C environment for clones isolated from evolution in either YPD 30°C or SC 37°C.

**Figure supplement 5.** Patterns of fitness-correlated epistasis with clones treated separately.

after a Benjamini–Hochberg multiple-hypothesis correction, these values fall to 24/77 (~31%), 15/74 (~20%), 9/77 (~12%), and 11/74 (~15%), respectively.

## Modeling the determinants of fitness effects

The examples in *Figure 2* demonstrate that correlations between fitness effects and fitness are common but often do not explain the bulk of epistasis. By definition, the fitness-correlated patterns we observe are the result of interactions between our insertion mutations and mutations that fix during the evolution experiment. If these interactions are all strictly 'fitness-mediated,' the fitness effects of mutations will be fully explained by a background fitness effect. Alternately, correlations between fitness effects and fitness could arise based on the average effect of a number of idiosyncratic effects that are more likely to be negative versus positive. To understand the relative contribution of these determinants of epistasis, we compare three linear models used to explain the fitness effects of a single mutation in each of our two environments:

1. The fitness model (XM): the fitness effect of the mutation is a linear function of background fitness.
2. The idiosyncratic model (IM): the fitness effect of the mutation can change in any population at any timepoint (and all subsequent timepoints) when an interacting mutation fixes in that population (see below for our constraints on fitting these parameters).
3. The full model (FM): the fitness effect of the mutation is affected by both a linear effect of background fitness and the idiosyncratic interactions of fixed mutations, as in the idiosyncratic model.

We fit each model by ordinary least squares (OLS). We define the fitness effect of each mutation in the ancestral strain as the mean fitness effect measured among clones from the first timepoint, and fix the intercept of each of our models accordingly. For the idiosyncratic and full models, we add idiosyncratic parameters iteratively, choosing the parameter that improves the Bayesian information criteria (BIC) the most at each step. These coefficients represent epistasis between the insertion mutation in question and one or more unknown mutations that fix during evolution in one population. Because mutations generally fix between every pair of timepoints during evolution, there could in principle be one idiosyncratic parameter for each timepoint in each population, but allowing all of these parameters would constitute overfitting. To combat this possibility, we do not allow parameters that fit a single point (e.g., a parameter for an effect at the final timepoint), and we only allow one parameter per population. We stop this iterative process of adding parameters if the BIC improves by less than 2 during a step (or when there is one parameter per population). Note that because of this iterative parameter adding procedure, the full model may have a different set of idiosyncratic parameters and will sometimes have less explanatory power than the idiosyncratic model (i.e., IM is not nested in FM).

*Figure 3A* shows how well each model explains the fitness effect data for each mutation in each environment. We find that the fitness model often explains a modest amount of variance (mean $R^2$: 18% in YPD 30°C, 19% in SC 37°C), but the idiosyncratic model (mean $R^2$: 54% in YPD 30°C, 41% in SC 37°C) and the full model (mean $R^2$: 53% in YPD 30°C, 44% in SC 37°C) usually offer more explanatory power. The examples in *Figure 3B* also demonstrate that epistasis is not strictly fitness-mediated; we commonly observe a stepwise change in the fitness effect of an insertion mutation in one evolving population, likely indicating epistasis between the insertion mutation and one or more mutations that fix in that population at a particular timepoint.

We can also ask which model best explains the data using the BIC, which penalizes models based on the number of parameters. The small squares below the bars in *Figure 3A* indicate which model has the lowest BIC for each mutation. In YPD 30°C, the full model has the lowest BIC for 40/77 (~52%) mutations and the idiosyncratic model has the lowest BIC for 37/77 (~48%). In SC 37°C, the full model has the lowest BIC for 49/73 (~67%) mutations and the idiosyncratic model has the lowest BIC for 24/73 (~33%). When we assess how well each model fits the entire dataset in each environment, the full model has a lower BIC than the idiosyncratic model in both environments.

Positive and negative coefficients in the idiosyncratic model represent positive and negative epistasis between mutations that fix during evolution and our insertion mutations. While these coefficients can arise in our modeling procedure due to noise alone, we find far more coefficients in our empirical dataset than in a simulated or shuffled dataset (see 'Materials and methods,' *Figure 3—figure supplement 4*). The bulk of the coefficients we find in YPD 30°C are negative (214/291, ~74%), while both

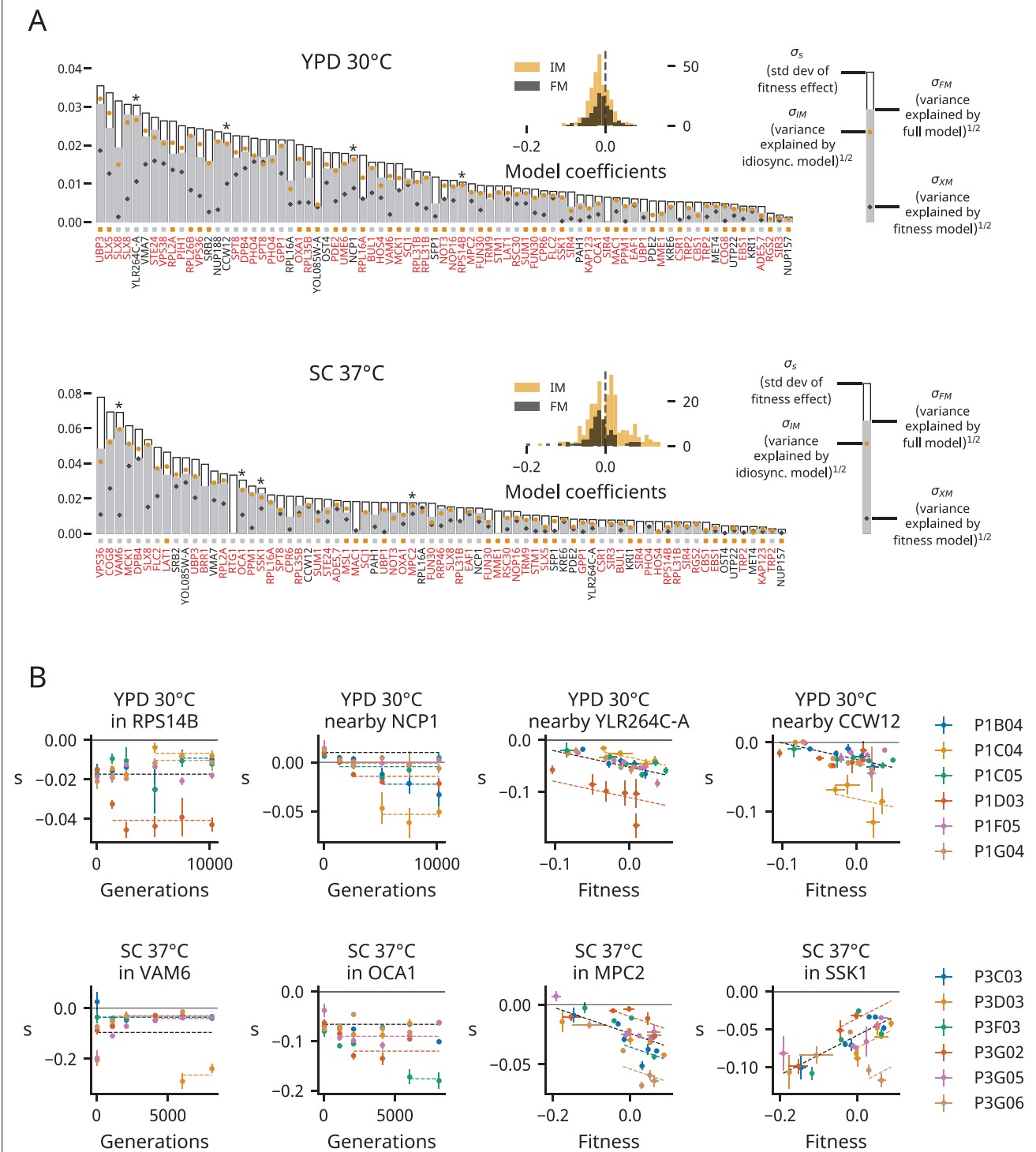

**Figure 3.** Determinants of fitness effects. (**A**) For each environment, we plot the standard deviation of the fitness effect across all population-timepoints and the square root of the variance explained by each of our three models. The colored squares below each bar represent which model has the lowest Bayesian information criteria (BIC) for each mutation. Mutations shown in red or black are insertions in or near the corresponding gene, respectively; stars indicate the mutations shown in panel (**B**). Only mutations with fitness-effect measurements in at least 20 population-timepoints are shown. The insets show the distribution of all coefficients in the idiosyncratic model (IM) and full model (FM), pooled across all mutations. (**B**) Examples of idiosyncratic (left half) and full (right half) model fits. Model predictions are shown by dashed lines, and lines with contributions from indicator variables associated with a particular population are the same color as the points from that population (colors are the same as in *Figure 1*).

The online version of this article includes the following figure supplement(s) for figure 3:

*Figure 3 continued on next page*

*Figure 3 continued*

**Figure supplement 1.** Epistasis in mutations that are beneficial on average at the first timepoint in at least one environment.

**Figure supplement 2.** Idiosyncratic model coefficients, broken down by population and timepoint in each condition.

**Figure supplement 3.** Model coefficients plotted by mutation.

**Figure supplement 4.** Model coefficient distributions for empirical, shuffled, and simulated datasets.

**Figure supplement 5.** Analogous to *Figure 3* but with clones treated separately.

**Figure supplement 6.** Determinants of fitness effects under the idiosyncratic model.

**Figure supplement 7.** Determinants of fitness effects under the idiosyncratic model.

**Figure supplement 8.** Determinants of fitness effects under the idiosyncratic model.

**Figure supplement 9.** Determinants of fitness effects under the idiosyncratic model.

**Figure supplement 10.** Determinants of fitness effects under the idiosyncratic model.

**Figure supplement 11.** Determinants of fitness effects under the idiosyncratic model.

**Figure supplement 12.** Determinants of fitness effects under the idiosyncratic model.

**Figure supplement 13.** Determinants of fitness effects under the idiosyncratic model.

**Figure supplement 14.** Determinants of fitness effects under the idiosyncratic model.

**Figure supplement 15.** Determinants of fitness effects under the idiosyncratic model.

**Figure supplement 16.** Determinants of fitness effects under the idiosyncratic model.

**Figure supplement 17.** Determinants of fitness effects under the idiosyncratic model.

**Figure supplement 18.** Determinants of fitness effects under the idiosyncratic model.

**Figure supplement 19.** Determinants of fitness effects under the idiosyncratic model.

**Figure supplement 20.** Determinants of fitness effects under the idiosyncratic model.

**Figure supplement 21.** Determinants of fitness effects under the idiosyncratic model.

**Figure supplement 22.** Determinants of fitness effects under the full model.

**Figure supplement 23.** Determinants of fitness effects under the full model.

**Figure supplement 24.** Determinants of fitness effects under the full model.

**Figure supplement 25.** Determinants of fitness effects under the full model.

**Figure supplement 26.** Determinants of fitness effects under the full model.

**Figure supplement 27.** Determinants of fitness effects under the full model.

**Figure supplement 28.** Determinants of fitness effects under the full model.

**Figure supplement 29.** Determinants of fitness effects under the full model.

**Figure supplement 30.** Determinants of fitness effects under the full model.

**Figure supplement 31.** Determinants of fitness effects under the full model.

**Figure supplement 32.** Determinants of fitness effects under the full model.

**Figure supplement 33.** Determinants of fitness effects under the full model.

**Figure supplement 34.** Determinants of fitness effects under the full model.

**Figure supplement 35.** Determinants of fitness effects under the full model.

**Figure supplement 36.** Determinants of fitness effects under the full model.

**Figure supplement 37.** Determinants of fitness effects under the full model.

positive (131/245, ~53%) and negative (114/245, ~47%) coefficients are common in our idiosyncratic model in SC 37°C (*Figure 3A* insets). Note that many of the positive epistatic terms in our idiosyncratic model in SC 37°C are the result of a consistent reduction in the fitness costs of some deleterious mutations in the first 2000 generations of evolution in all populations (e.g., see the mutation in VAM6 in *Figure 3B* and others in *Figure 3—figure supplements 6–37*, and see *Figure 3—figure supplement 3* for a breakdown of coefficients by individual mutations).

The differences we observe in epistatic patterns between environments could be caused by interactions between epistasis and the environment ('G×G×E' effects), differences in the adaptive targets (i.e., the functional modules subject to selection) in each environment, or a combination of the two. To tease apart these possibilities, we measured the fitness effects of mutations in clones evolved in YPD

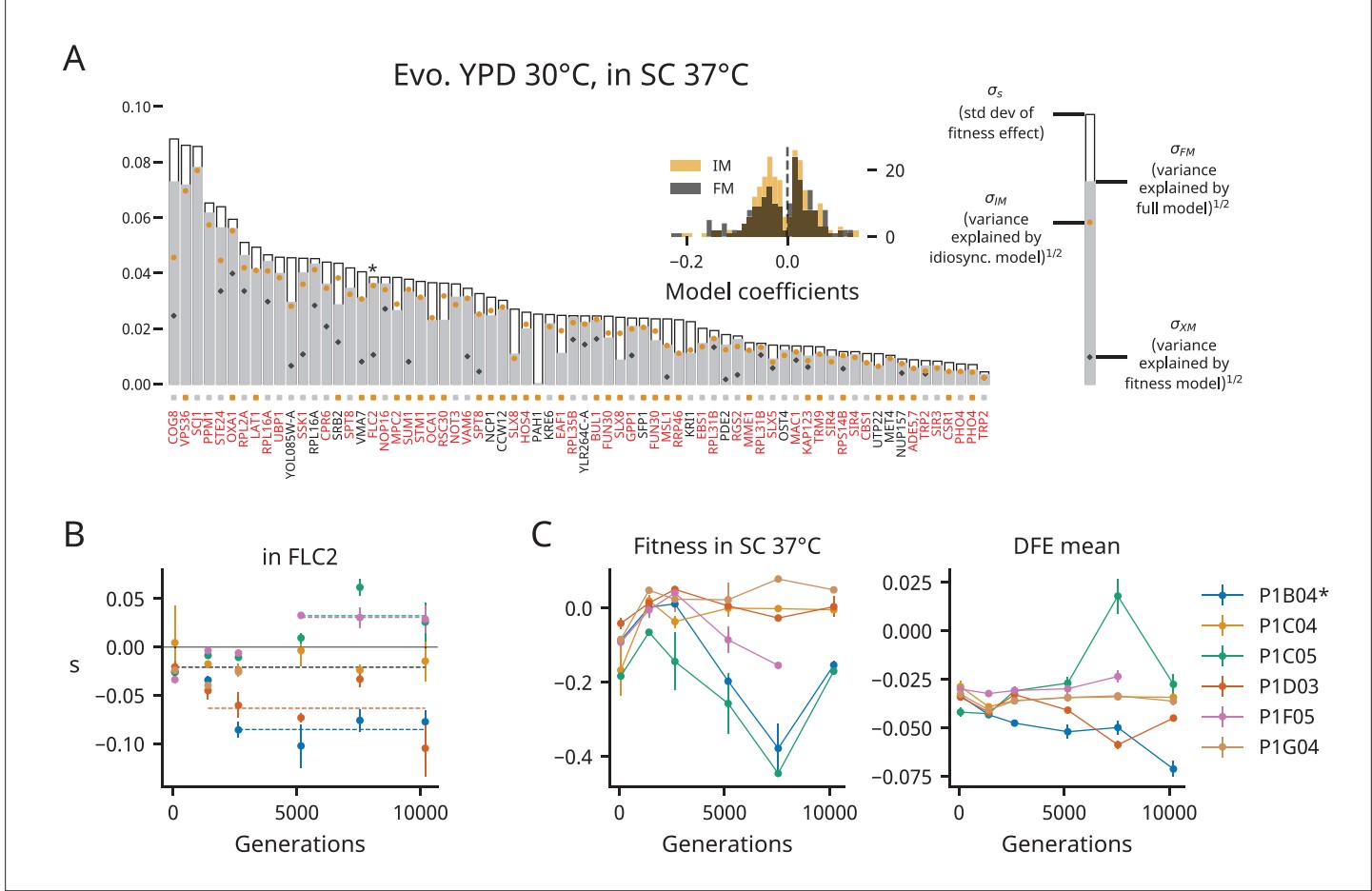

**Figure 4.** Patterns of epistasis in a nonevolution environment. (**A**) Same as *Figure 3A*, but for clones from YPD 30°C assayed in SC 37°C. We plot the standard deviation of the fitness effect across all population-timepoints and the square root of the variance explained by each of our three models. The colored squares below each bar represent which model has the lowest Bayesian information criteria (BIC) for each mutation. Mutations shown in red or black are insertions in or near the corresponding gene, respectively; stars indicate the mutations shown in panel (**B**). Only mutations with fitness-effect measurements in at least 20 population-timepoints are shown. The inset shows the distribution of all coefficients in the idiosyncratic model (IM) and full model (FM), pooled across all mutations. (**B**) Example IM model fit, as in *Figure 3B*. The model predictions are shown by bold dashed lines, and lines with contributions from indicator variables associated with a particular population are the same color as the points from that population (colors are the same as in *Figure 1* and panel **C**). (**C**) The fitness and distribution of fitness effects (DFE) mean over time in YPD 30°C populations assayed in SC 37°C. The asterisk indicates a significant correlation (p<0.05). Error bars on fitness represent the standard deviation of the fitnesses measured for the two clones, but note that we were only able to measure the fitness of one clone at several population-timepoints due to low fitnesses relative to our reference; the corresponding points here have no error bars. Error bars on the DFE mean represent the standard error of the DFE mean, calculated from the standard errors of individual mutations (see 'Materials and methods').

The online version of this article includes the following figure supplement(s) for figure 4:

**Figure supplement 1.** Analogous to *Figure 4* but with clones treated separately.

30°C in the SC 37°C environment. We assayed the background fitness of each of these clones in SC 37°C and found that populations evolved in YPD 30°C sometimes experience large fitness declines in the SC 37°C environment (*Figure 4C*). We also observe widespread epistasis in this alternate environment, but do not observe any overall trends in the mean of the DFE in SC 37°C over the course of evolution in YPD 30°C (*Figure 4A and B*). Instead, the variance in our data is dominated by one particularly low fitness clone (from population P1C05, generation 7550) in which several insertion mutations were strongly beneficial (*Figure 4C*).

When we apply the same set of models to this dataset, we again find that both background fitness and idiosyncratic effects can explain how many mutations' fitness effects vary, with the latter again outperforming the former (*Figure 4A*). Notably, we observe relatively more positive epistatic

coefficients (109/241, ~45%) than in YPD 30°C (77/291, ~26%), though we also observe a heavy tail of strongly negative coefficients. These patterns of idiosyncratic epistasis are not due to outsized contributions from a few populations; the distributions of IM coefficients for YPD 30°C populations are different in the two assay environments across populations (*Figure 3—figure supplement 2*). Overall, these results support the hypothesis that G×G×E effects underlie the differences we observe between environments (see also *Hall et al., 2019*), though they do not rule out the possibility that differences in adaptive targets between environments may also contribute.

## Discussion

### Shifting distributions of fitness effects

By using our barcode-based mutagenesis system to assay the fitness effects of 91 specific gene disruption mutations across numerous genetic backgrounds spanning 8000–10,000 generations of laboratory evolution, we have described how overall mutational robustness (defined in terms of the average effect of this type of insertion mutation) changes during evolution. We then dissected these overall effects in terms of how the fitness effects of individual mutations change during evolution. We find that populations adapting to our YPD 30°C environment become less robust to deleterious mutations over time. This shift in the mean of the DFE in YPD 30°C is not caused by strictly fitness-mediated shifts in the fitness effects of mutations, but is instead the result of an excess of negative idiosyncratic epistatic effects (*Figure 3*). In contrast, in clones isolated from populations evolved in SC 37°C, epistatic interactions are more evenly divided between negative and positive effects.

The fact that populations evolved in YPD 30°C lost robustness as they increased in fitness over time is broadly consistent with our earlier work showing that the DFE becomes more strongly deleterious in more-fit genetic backgrounds (*Johnson et al., 2019*). However, the loss of robustness we observe here was not as strong as in this previous work (see *Figure 1—figure supplement 6* for a comparison to the effect size in *Johnson et al., 2019*), and we did not observe any predictable change in the DFE mean in SC 37°C. One potential explanation for the weaker patterns we observe at the DFE level in this experiment is that there are simply less mutations involved compared to the genetic backgrounds from our earlier work: the genotypes in our previous experiment were derived from a cross between two yeast strains that differed at tens of thousands of loci, so the pool of mutations was both larger and more balanced (each mutation present in ~50% of clones) than in this study. If the overall patterns of fitness-correlated epistasis arise due to the collective effect of numerous idiosyncratic interactions, rather than a genuine fitness-mediated effect, we would therefore expect weaker trends here (*Lyons et al., 2020*; *Reddy and Desai, 2021*).

Our results also illustrate a form of hidden evolutionary unpredictability, despite the fact that our populations increased in fitness over time along a predictable trajectory (*Johnson et al., 2021*). As these populations adapted, they accumulated mutations that carry with them epistatic interactions with potential future mutations across the rest of the genome. The patterns of epistasis we observe among the insertion mutations in our study demonstrate that the predictability of these second-order effects varies widely: some potential future mutations show strong fitness-correlated effects in every population, while others are affected by a small number of idiosyncratic interactions with mutations that fix during evolution. These kinds of unpredictable patterns of epistasis could lead to fundamentally unpredictable evolutionary outcomes, with changes in the fitness effects of mutations dynamically closing off or opening up evolutionary pathways during adaptation.

### Patterns of epistasis among individual mutations

Most of the insertion mutations we analyze in this experiment are deleterious across most or all genetic backgrounds. We find that these deleterious mutations tend to become more strongly deleterious over time in populations evolved in the YPD 30°C environment. This pattern results from an overabundance of negative epistatic interactions, which could involve either the beneficial mutations that drive fixation events or the neutral or weakly deleterious mutations that hitchhike to fixation during selective sweeps (*McDonald et al., 2016*). The strong predictive power of background fitness for the fitness effects of some mutations suggests that interactions with beneficial mutations are driving these patterns of epistasis, but idiosyncratic effects that deviate from these relationships hint at a role for interactions with hitchhiker mutations as well (*Figure 3*). Systematic backcrossing and mutagenesis

experiments would be required to disentangle these patterns (along with any cases of higher-order epistasis involving multiple mutations that fix during evolution), but we suspect both types of interactions contribute to the epistasis we observe.

Among the relatively small number of beneficial insertion mutations we analyze, we find that the beneficial effects tend to decline over time, and almost universally shift to neutral or deleterious effects later in the experiment (*Figure 3—figure supplement 1*). These results provide additional examples of diminishing-returns epistasis among beneficial mutations, which can at least partially explain the pattern of declining adaptability often observed in microbial evolution experiments (*Chou et al., 2011*; *Khan et al., 2011*; *Kryazhimskiy et al., 2014*; *Wünsche et al., 2017*). One of these mutations has a clear functional story behind it: the beneficial effect of an insertion mutation in ADE5,7 is the result of breaking the adenine synthesis pathway upstream of a toxic intermediate, so when populations in SC 37°C fix other loss-of-function mutations that also break the pathway between the first two sampled timepoints in the evolution experiment, this mutation becomes neutral (we do not see this effect in YPD 30°C because most populations have fixed a mutation in the adenine pathway by the first sampled timepoint; for a more in-depth discussion of this case of epistasis and contingency, see *Johnson et al., 2021*).

The picture that emerges from our data is one in which idiosyncratic epistatic effects are largely unpredictable but also unbalanced (have a nonzero mean), which leads to correlations with background fitness. In both our own work and other studies of microbial evolution, the mean epistatic effect is negative: beneficial mutations tend to have negative epistatic interactions with both deleterious (*Johnson et al., 2019*, this study) and beneficial mutations (*Chou et al., 2011*; *Hall et al., 2019*; *Karkare et al., 2021*; *Khan et al., 2011*; *Kryazhimskiy et al., 2014*; *Ono et al., 2017*; *Pearson et al., 2012*; *Perfeito et al., 2014*; *Rokyta et al., 2011*; *Wünsche et al., 2017*).

At the broadest level, we can distinguish two potential sources of these unbalanced patterns of epistasis: the structure of biological systems and the set of mutations that fix during evolution. Both theoretical and experimental work has shown that genes within functional modules tend to have similar interaction profiles with other genes (*Costanzo et al., 2016*; *Segrè et al., 2005*). Given this kind of 'monochromatic' epistasis, all that is necessary for fitness-correlated epistasis to appear during evolution is for beneficial mutations to be clustered in a few modules. The strength and direction of fitness-correlated epistasis will then depend on the particular modules targeted by selection and how those modules interact with the rest of the cell. For example, the targets of adaptation in SC 37°C may be more related to heat stress than general components of growth (e.g., mutations in LCB3, which have been shown to reduce killing in certain heat stress conditions, are enriched in populations in SC 37°C) (*Ferguson-Yankey et al., 2002*; *Johnson et al., 2021*). If the strongly deleterious effects of some of our insertion mutations are exacerbated by heat stress, but a beneficial mutation reduces heat stress, we can expect a positive interaction between the two mutations. In contrast, we hypothesize that the deleterious effects of some insertion mutations become more pronounced when growth rate is increased by mutations in YPD 30°C (*Johnson et al., 2019*). In order to understand or predict these differences in epistasis specific to the evolution environment, we will need to better understand the functional structure of biological systems.

## Higher-order epistasis and the evolution of robustness

Each mutation that fixes during evolution has an immediate first-order effect on fitness, but also carries with it a second-order set of pairwise interactions whose strength and direction is determined by the structure of functional relationships between genes and biological modules. Through higher-order interactions, a mutation can also change the structure of these functional relationships, altering the complexity, redundancy, robustness, and evolvability of biological systems (reviewed in *de Visser et al., 2003*; *Masel and Siegal, 2009*; *Masel and Trotter, 2010*; *Payne and Wagner, 2019*). Our work here suggests that mutational robustness tends to decrease during evolution in some environments, but our data is limited to interactions with the set of ~200 mutations that fixed during 10,000 generations of laboratory adaptation. We expect the effects we observe here to dominate during rapid adaptation, but over longer evolutionary timescales, robustness may be more dependent on changes in the higher-order structure of biological systems.

## Ideas and speculation

We find that in one evolution environment a tendency towards negative epistasis leads to a shift in the DFE towards more deleterious mutations over time, while in another evolution environment no such shift occurs. Clearly, the details matter, and it is difficult to draw general conclusions. In this section, we speculate on how work in metabolic control theory (MCT) may help explain the functional underpinnings of our results and fitness-correlated epistasis more generally.

MCT describes how mutations in idealized metabolic pathways change the *control coefficients* of other mutations (*Szathmáry, 1993*). In a strictly serial pathway, a mutation reducing the activity of one enzyme will decrease the control other enzymes have over flux through the pathway, but a mutation increasing the activity of one of the enzymes will increase the control of others. If two enzymes act instead in parallel, these patterns are reversed: a mutation that increases activity of one will decrease the control of the other on flux, and vice versa (for a thorough treatment of how epistasis arises and propagates in metabolic networks, see *Kryazhimskiy, 2021*).

If flux through the pathway is correlated with fitness, these patterns of interactions predict epistasis between mutations in different enzymes in the pathway. For example, if flux is *negatively* correlated with fitness in a strictly serial pathway, beneficial mutations will be those that reduce flux, so they will reduce the control of other enzymes in the pathway, such that these beneficial mutations exhibit negative epistasis. This is the case for beneficial loss-of-function mutations in the broken adenine-synthesis pathway present in the ancestors of our evolution experiment (*Johnson et al., 2021*): one beneficial loss-of-function mutation in this pathway (e.g., in ADE4) lowers the control coefficients of the rest of the enzymes in that pathway, such that loss-of-function mutations in these enzymes become less beneficial (e.g., our insertion mutation in ADE5,7 in this experiment, *Figure 3—figure supplement 1*).

MCT is a mathematical framework for describing these interactions, but the general qualitative principles can be applied beyond enzymes in metabolic pathways to understand patterns of fitness-correlated epistasis (*MacLean, 2010*). One hypothesis for the functional underpinnings of increasing-costs epistasis can be framed in this way. First, we assert that the large-scale components of *growth* in the yeast cell *functionally* act in serial. While these components of growth (e.g., cell wall production, ribosome production, and DNA replication) do not belong to an actual serial pathway, they clearly do not act in parallel: these components generally cannot 'fill in' for each other. That is, they are *nonredundant*. We therefore propose that an increase in the function of one of these components (in terms of growth rate) will increase the control coefficients of the rest of the components. Similarly, a decrease in the function of one component will decrease the control coefficients of the rest. We can intuitively understand this based on the idea of limitation: if DNA replication slows to a halt, growth rate will become less sensitive to changes in the speed of cell wall production. In contrast, if a population in which growth is limited by DNA replication fixes a mutation that improves DNA replication, we expect the control coefficient of cell wall production to increase, meaning deleterious mutations that slow cell well production will become more deleterious. We believe this effect can explain much of the increasing-costs epistasis we have observed.

Consider the following metaphor for this effect. You work for a car manufacturer, and your factory's goal is to produce cars as quickly as possible. You work with a small team that builds the wheels. Your team is efficient, but the engine team is much slower. Because the engine team is limiting production, you don't feel under pressure from the boss at all – in fact, one of your team members slacks off sometimes, but the company hardly suffers (read: their control coefficient is low, deleterious mutations have small effects). One day the engine team purchases a new robot and dramatically speeds up their process. Suddenly cars are waiting for wheels, and the pressure on your team increases dramatically – now when your teammate is slacking, it slows the entire production line down (read: their control coefficient is high, deleterious mutations have larger effects, costs have increased).

In our discussion, we speculate that we see increasing costs more frequently in YPD 30°C because adaptation in that environment is more focused on improving core components of growth compared to adaptation in SC 37°C where selection for improvement in heat tolerance or survival may be more common. The phenotype of survival does not fit as neatly into our car manufacturing metaphor: we have no strong hypotheses for how control coefficients should change as populations increase heat tolerance or for how large-scale phenotypes such as growth and survival integrate in terms of competitive fitness. However, we speculate that in SC 37°C adaptive mutations are less likely to be affecting the core components of growth, and therefore cause increasing costs less often than mutations

selected in YPD 30°C (i.e., adaptive mutations in SC 37°C are less likely to increase the control coefficients of random mutations). To move closer to predictive models of epistasis, we will need to better understand the functional relationships between these large-scale cellular phenotypes.

We end our discussion by noting that changes in robustness over longer evolutionary timescales may depend not on the type of changes we observe in our evolution experiment, but on changes to the higher-order structure of biological systems. Within our metaphor, this is the difference between changes to each team's effectiveness and changes to the structure of the assembly line (or even to the ultimate product of the system). Defining what constitutes a change to the structure of the system itself is a difficult problem, analogous to defining 'novelty' in evolution (*Murray, 2020*), but most would agree that there are fundamental differences in the complexity and redundancy of the genetic systems of viruses and humans, for example.

Both experimental and theoretical work have suggested that higher genome complexity and redundancy is associated with more negative epistasis between deleterious mutations (*Macía et al., 2012*; *Sanjuán and Elena, 2006*; *Sanjuán and Nebot, 2008*; though see *Agrawal and Whitlock, 2010*). Negative epistasis between deleterious mutations implies that deleterious mutations increase the control coefficients of other mutations and that beneficial mutations will decrease control coefficients. We can understand this pattern as a broad extension of MCT: the case of strictly parallel pathways and fitness correlated with flux, in which two deleterious mutations exhibit negative epistasis, is one example of a functionally redundant relationship. Therefore, in organisms with higher functional redundancy, we expect to observe more positive interactions between beneficial and deleterious mutations and less increasing-costs epistasis. In other words, the second-order loss of mutational robustness we observe during adaptation should be stronger in organisms with low redundancy. While there may be more unseen factors affecting the types of interactions beneficial mutations participate in, present data suggests that increasing-costs epistasis may be specific to organisms or environments where the functional redundancy of genes or biological modules is low.

## Epistasis between beneficial mutations

An apparent contradiction emerges from our explanation for increasing-costs epistasis: if adaptation increases the control coefficients of core components of growth, why do we not see more positive epistasis between beneficial mutations in evolution experiments? Why do we instead usually see negative, diminishing-returns epistasis and declining adaptability? We propose that this discrepancy arises due to differences in the availability and form of beneficial and deleterious mutations. Based on previous work, we expect beneficial mutations in laboratory evolution experiments to be primarily loss-of-function mutations (*Murray, 2020*). In contrast to deleterious mutations, which can be spread across the genome, these types of beneficial mutations will rarely exist in well-adapted core components of growth, and will instead be clustered in a few adaptive targets. Within these targets, we believe loss-of-function beneficial mutations are often *functionally redundant*, meaning that they tend to *decrease* each other's control coefficients for fitness. Beneficial mutations can be redundant by inactivating the same deleterious pathway (e.g., ADE pathway mutations discussed above), solving the same general problem (e.g., mutations shortening lag in *Karkare et al., 2021*), or changing a phenotype with a nonlinear fitness function (*Chiu et al., 2012*; *Chou et al., 2014*; *Keren et al., 2016*; *Lunzer et al., 2005*; *Otwinowski et al., 2018*). Nonmonotonic fitness functions can arise from phenotypes with both potential benefits and costs (*Dekel and Alon, 2005*), such that negative interactions between beneficial mutations and the benefit can lead to fitness-correlated epistasis that crosses neutrality, exhibiting diminishing returns, increasing costs, and sign epistasis (*Figure 3—figure supplement 1*).

This explanation provides a prediction: we will be more likely to see synergistic epistasis during evolution experiments when we observe beneficial *gain-of-function* mutations. *Chou et al., 2009* provides a particularly strong example of a beneficial gain-of-function (promoter capture) mutation that is more beneficial in more-fit genetic backgrounds. In the long-term *Escherichia coli* evolution experiment, potentiating mutations acquired during adaptation in one population interacted positively with a beneficial gain-of-function mutation (also a promoter capture), enabling aerobic citrate utilization (*Blount et al., 2012*). Studies of evolutionary repair also provide examples of synergistic interactions between apparently nonredundant beneficial mutations (*Fumasoni and Murray, 2020*; *Hsieh et al., 2020*). These counterexamples underscore the fact that diminishing-returns epistasis is not a rule; it is a pattern that is overrepresented in evolution experiments due to a tendency for

beneficial mutations to be loss-of-function mutations and to be clustered in a few adaptive targets. These tendencies may be weaker later in evolution experiments when mutations are spread more evenly across cellular modules, such that a period of declining adaptability caused by diminishing-returns epistasis early in an experiment gives way to a period of relatively constant fitness gains (*Good and Desai, 2015*).

## A final note on terminology for epistasis

Very few papers discuss epistasis between beneficial and deleterious mutations – most theoretical and experimental work has focused on epistasis between two beneficial mutations or two deleterious mutations. With these same-signed pairs of mutations, the terms negative and positive epistasis are consistent. Two beneficial mutations that interact negatively imply two deleterious reversions that also interact negatively. However, when we consider one of these beneficial mutations and the deleterious reversion of the other, they interact positively. We provide this note to clarify that increasing-costs epistasis, in which deleterious mutations exhibit negative epistasis with beneficial mutations, should not be considered the same as previous results demonstrating negative epistasis between two deleterious or two beneficial mutations. Instead, we should expect it in systems where more positive epistasis is observed between same-signed mutations. While epistasis is already a concept overladen with terminology, we submit that in some cases it may be more useful to classify interactions between mutations or cellular components as being functionally redundant or nonredundant in terms of fitness.

# Materials and methods

## Strains

All strains used for this study were isolated from the evolution experiment described in *Johnson et al., 2021*. We isolated two clones from each of our focal populations at each sequencing time-point. For this experiment, we used clones from 12 MATa populations, 6 from the YPD 30°C environment and 6 from the SC 37°C environment. We decided to include population P1B04, which exhibits a cell-clumping phenotype in preliminary imaging data, and to exclude population P1B03, which diploidized during evolution, and populations P3C04, P3F05, P3D05, and P3E02, which lost G-418 resistance during evolution (not being able to select on G-418 during transformation could allow the HygMX cassette to replace the KanMX cassette, leading to leakage during the selection step). Otherwise, we chose populations randomly. The ancestor of these populations is MJM361 (MATa, YCR043C:KanMX, STE5pr-URA3, ade2-1, his3Δ::3xHA, leu2Δ::3xHA, trp1-1, can1::STE2pr-HIS3 STE3pr-LEU2, HML::NATMX, rad5-535).

## Barcoded Tn7 libraries

We used a previously created set of Tn7-based plasmid libraries to introduce the same set of ~100 mutations into each of our strains (*Johnson et al., 2019*). These plasmids contain a section of the yeast genome corresponding to one of these ~100 locations, interrupted by a Tn7 insertion containing a random DNA barcode and a HygMX cassette. Each barcode uniquely identifies the mutation and the plasmid library via a mapping established in earlier work (*Johnson et al., 2019*).

## Transformation

Our yeast transformation protocol is a scaled-up version of that used in *Johnson et al., 2019*, based on the method described in *Gietz and Schiestl, 2007*. We grew strains from freezer stocks overnight, diluted 750 μL into 15 mL YPD + ampicillin (100 g/mL), grew for 4 hr, pelleted the cells, and resuspended in 900 μL transformation mix and 100 μL plasmid DNA cut with NotI-HF (corresponds to 2 μg of plasmid; cut at 37°C for 3 hr, then heat-inactivated at 65°C for 10 min). We then heat-shocked this mixture at 42°C for 1 hr, recovered in 3 mL YPD + ampicillin for 2 hr, plated 25 μL on antibiotic selection plates to check efficiency, and then combined the rest with 40 mL YPD supplemented with antibiotics. For both agar and liquid-selective media, we included hygromycin (300 μg/mL), clonNat (20 μg/mL), and G-418 (200 μg/mL). We made frozen glycerol stocks of each transformation after ~48 hr of growth. All growth was conducted at 30°C either in a test tube on a roller drum (recovery) or in a baffled flask on an orbital shaker (all other steps).

We transformed 2 clones from 12 populations at 6 timepoints for a total of 144 transformations. We organized these transformations into three 'VTn assays,' each associated with 48 transformations using our 48 unique barcoded libraries.

## Fitness assays

Again, we followed the protocols established in *Johnson et al., 2019* for our fitness assays. We used two types of media: rich YPD media (1% Bacto yeast extract [VWR #90000-726], 2% Bacto peptone [VWR #90000-368], 2% dextrose [VWR #90000-904]) and synthetic complete (SC) media (0.671% YNB with nitrogen [Sunrise Science #1501-250], 0.2% SC [Sunrise Science # 1300-030], 2% dextrose). We assayed our transformed libraries of clones from the YPD 30°C environment in both their evolution environment (YPD 30°C) and the SC 37°C environment, and clones from the SC 37°C environment in their evolution environment (SC 37°C). We first arrayed our transformation glycerol stocks into two 96-well plates corresponding to the two evolution environments, and then inoculated 8 µL from each well of these plates into four (for SC 37°C assays) or eight (for YPD 30°C assays) flat-bottom polypropylene 96-well plates containing 126 µL of media, supplemented with the same antibiotics as during the initial selection. To ensure efficacy of the antibiotics in the SC 37°C environment, we used media with MSG instead of ammonium sulfate (1.71 g/L YNB without amino acids or ammonium sulfate, 2 g/L SC, 1 g/L MSG). After this period of growth, we used YPD and SC supplemented with ampicillin (100 µg/mL) and tetracycline (25 µg/mL), matching the conditions of the evolution experiment. After 40 hr of growth in these plates, we started daily transfers.

At each daily transfer, we diluted YPD 30°C cultures $1/2^{10}$ and SC 37°C cultures $1/2^{8}$. During these transfers, we combined and mixed cultures from each well corresponding to the same clone/transformation to increase population size and reduce bottleneck noise. In the first (T0) transfer, we combined cultures from the eight plates that were initially inoculated from the freezer stock and diluted them into 20 96-well plates. In all subsequent transfers (T1–4), we combined cultures from all 20 plates and diluted them into 20 new plates. Specifically, for YPD 30°C, we diluted 3 µL from each well of 20 plates into 60 µL YPD (60 µL total, 1/2 dilution), mixed, then diluted 16 µL into 112 µL YPD ($1/2^{3}$ dilution), mixed, and distributed 2 µL into 126 µL YPD in 20 plates ($1/2^{6}$ dilution). For SC 37°C, we diluted 3 µL from each well of 20 plates into 60 µL SC (60 µL total, 1/2 dilution), mixed, then diluted 60 µL into 60 µL SC (1/2 dilution), mixed, and distributed 2 µL into 126 µL SC in 20 plates ($1/2^{6}$ dilution).

## Barcode sequencing

Our fitness assays in YPD 30°C were originally performed alongside assays in SC 37°C that were later abandoned due to an issue with expired reagents (and repeated with appropriate reagents in our second round of assays). During these assays, we combined equal volumes of culture at the end of each transfer from every well corresponding to each of the three VTn assays in each environment. We can pool the cultures corresponding to each VTn assay because we know which barcodes correspond to which plasmid library/clone, so we can divide our barcode count data appropriately during sequencing analysis. We performed DNA extractions from two 1.5 mL pellets for each assay-timepoint from our YPD 30°C fitness assays and from four 1.5 mL pellets from our SC 37°C fitness assays using Protocol I from the Yeastar Genomic DNA Kit (Zymo Research), as described previously (*Johnson et al., 2019*). We then amplified barcodes using a two-step PCR protocol. We performed four first-round PCRs with 19 µL gDNA, 25 L 2X Kapa Hotstart Hifi MM, 3 µL 10 M TnRS1 primer, and 3 µL 10 M TnFX primer, and ran the PCR protocol: (1) 95°C 3:00, (2) 98°C 0:20, (3) 60°C 0:30, (4) 72°C 0:30, GO TO step 2 three times, and (5) 72°C 1:00. We purified these PCRs with PCRClean DX Magnetic Beads (Aline) using a 0.85× ratio. We then set up two second-round PCRs per sample by combining 25 µL purified PCR µ1 product, 1.5 µL ddH$_2$O, 10 µL Kapa Hifi Buffer, 1 µL KAPA HiFi HotStart DNA Polymerase, 5 µL 5 M N7XX primer (Nextera), and 5 µL 5 M S5XX primer (Nextera), and ran the PCR protocol: (1) 95°C 3:00, (2) 98°C 0:20, (3) 61°C 0:30, (4) 72°C 0:30, GO TO step 2 19 times, and (5) 72°C 2:00. We purified the resulting libraries with Aline beads, using a 0.7× ratio, then repeated the purification with a 0.65× ratio, and finally sequenced our pooled libraries on a NextSeq 550 (Illumina).

## From reads to barcode counts

We process our sequencing data as described previously (*Johnson et al., 2019*). We first filter reads based on inline indices and quality scores, use regular expressions to extract barcode sequences, and

combine barcode counts across timepoints for each VTn assay. Next, we use a single-bp-deletion-neighborhood method to correct errors in raw barcodes, assigning them to the set of known barcodes from each of our plasmid libraries. By associating barcodes with plasmid libraries, we associate them both with a fitness assay for a particular clone and with a particular insertion mutation, and we divide our barcode count data accordingly.

## Estimating fitness effects from barcode counts

Again, we follow *Johnson et al., 2019*, with minor differences. First, we convert barcode counts to log-frequencies at each timepoint. After this preliminary step, we noticed a large number of log-frequency spikes, restricted largely to one timepoint in one of our VTn assays in SC 37C. These spikes in frequency very likely represent low-level sequencing library contamination from another timepoint due to primer cross-contamination. In *Figure 1—figure supplement 9*, we show that this is confined to this single timepoint and demonstrate how we can use a simple heuristic (excluding lineages whose log-frequency at timepoint 2 is 0.5 greater than both timepoint 1 and timepoint 3) to remove the barcoded lineages affected by this sequencing library contamination. This step excludes, on average, less than 1% of the reads from timepoint 2 in these assays. Next, we calculate fitness effects for each barcode, as described in *Johnson et al., 2019*. After excluding timepoints with less than 5000 total barcode counts, we measure the log-frequency slope for each barcode at each consecutive pair of timepoints, excluding timepoints in which the barcode has less than 10 counts. We scale each of these log-frequency slopes by the median log-frequency slope of barcodes associated with five neutral reference mutations, and then average these scaled values to get one fitness measurement for each barcode. As in *Johnson et al., 2019*, we observe a small fraction of outlier barcodes, which follow starkly different log-frequency trajectories than the other barcodes associated with the same insertion mutation, presumably due either to pre-existing mutations in the transformed culture or transformation artifacts (including two mutations being transformed together). We use a log-likelihood ratio test to identify barcodes whose read counts are inconsistent with barcodes near the median fitness measured for one insertion mutation. Based on iterative exclusion and exploration of frequency trajectories for this experiment, we chose a heuristic cutoff of 40 for the log-likelihood ratio required to exclude barcodes (for a detailed description of this method, see *Johnson et al., 2019*). Finally, to decrease noise from low-frequency barcode lineages while retaining the independent measurements unique barcodes provide, we randomly combine counts from individual barcodes into a maximum of five combined barcodes ('cBCs') per insertion mutation. Next, we repeat the fitness measurement process described above to get final fitness measurements for each cBC ($s_{cbc}$).

Again following *Johnson et al., 2019*, we calculate the mean and standard error of all cBCs fitness measurements for each insertion mutation ($m$) in each clone ($c$):

$$S_{m,c} = \frac{\sum_{cbcs} s_{cbc}}{num\_cbcs}, \sigma_{m,c,cbc} = \sqrt{\frac{\sum_{cbcs} \left(s_{cbc} - s_{m,c}\right)^2}{\left(num\_cbcs - 1\right) * num\_cbcs}} ,$$

For each clone, we also calculate the standard error of our fitness measurements of our set of neutral barcodes ($\sigma_{neut,c}$). Since $s_{m,c}$ is the difference between the measured cBC fitness for mutation $m$ and the measurement of neutrality, we calculate the standard error for $s_{m,c}$ as the square root of the sum of squares of these two errors:

$$\sigma_{m,c} = \sqrt{\sigma_{m,c,cbc}^2 + \sigma_{neut,c}^2}$$

To obtain the fitness effect of a mutation for a population-timepoint ($s_{m,p,t}$), we calculate an inverse-variance weighted average on the fitness effect measurements from each clone:

$$s_{m,p,t} = \frac{\sum_{clones} \frac{s_{m,c}}{\sigma_{m,c}^2}}{\sum_{clones} \frac{1}{\sigma_{m,c}^2}}$$

We only calculate $s_{m,p,t}$ if there are at least three cBCs between the two clones. If one of the two clones has only one cBC, we use $\sigma_{m,c}$ from the other replicate as an estimate of the standard error to be used in inverse-variance weighted averaging. Note that because we only consider data with at

least three cBCs during our analysis in which we treat clones independently, this estimate of the standard error is only used for the inverse-variance weighted averaging.

We calculate the standard error on this $s_{m,pt}$ measurement as described above for individual clones using the $s$ values for all cBCs associated with mutation $m$ in both clones:

$$\sigma_{m,pt,cbc} = \sqrt{\frac{\sum_{cbc}\left(s_{cbc}-s_{m,p,t}\right)^2}{(num\_cbcs-1)*num\_cbcs}}$$

As described in *Johnson et al., 2019*, we use this conservative measure of standard error instead of the one given by inverse-variance weighting in order to better capture unspecified biological error between the two replicates. Again, we combine this standard error with the standard error of all neutral cBCs in both clones ($\sigma_{neut,pt}$):

$$\sigma_{m,p,t} = \sqrt{\sigma^2_{m,p,t,cbc} + \sigma^2_{neut,p,t}}$$

To test whether mutations have effects significantly different from zero for each population-timepoint, we fit all cBC $s$ values for a single mutation by OLS as described in *Johnson et al., 2019*. The OLS model includes a fitted term for the fitness effect of the mutation ($s_{mut}$), a fitted term ($\beta_{mut}$) for differences in the effect between clones (indicated by $r$), and a normally distributed noise term ($e_{mut}$).

$$s_{cbc} = s_{mut} + \beta_{mut}r + e_{mut}$$

We use the $t$-statistic for the intercept to calculate p-values for whether $s_{mut} \neq 0$ for each mutation. We perform a Benjamini–Hochberg correction at the 0.05 level on the entire set of p-values we find.

## Measuring changes in the DFE and accounting for missing measurements

For each population-timepoint or clone, in each condition, we calculate the mean of the fitness effects of all mutations with at least three cBCs, excluding distributions with less than 60 mutations with fitness effect measurements. We calculate the standard error of the mean of the distribution of fitness effects by combining variances from two sources: error in our measurement of mutation fitness effects and errors in our measurement of mean fitness. Since this second component of variance is shared across mutations, it is not scaled by the number of mutations:

$$\sigma_{DFE_{mean},p,t} = \sqrt{\frac{\sum_{mutations} \sigma^2_{m,p,t,cbc}}{n^2} + \sigma^2_{neut,p,t}}$$

where $n$ is the number of mutations with measurements in the distribution (we use an analogous formula in our analysis on individual clones). Importantly, note that this measure of error for the DFE mean describes only noise from measured mutations and does not address the fact that some mutations do not have measurements in some clones or population-timepoints.

We examined the effect of these missing fitness effect measurements in several ways. First, we examined the mean of the DFE in sets of mutations shared across the set of population-timepoints we had assayed successfully in each condition. For the analysis of individual clone DFEs in each condition, we try to find the largest set of clones with at least 40 mutations with measurements in every clone. We do this by iterating through the list of clones, sorted by the number of measured mutations, and adding them to a set of clones until the number of mutations with measurements in every clone drops below 40. Next, we create 'filled-in' DFEs for each population-timepoint and each clone in which missing measurements are replaced with the mean fitness effect measured across all strains in a given condition. The pattern observed in YPD 30°C, in which the mean fitness effect decreases as populations evolve and gain fitness, is stronger (in terms of the p-value and $R^2$ of the regression) in both the set of shared mutations and the filled-in DFE than in our original analysis.

Finally, we examined the number of strongly deleterious mutations (defined as having a mean fitness effect <–0.05 across all population-timepoints in a given condition) that were not measured in each population-timepoint. In YPD 30°C, these strongly deleterious mutations are less likely to be measured in more fit strains, again suggesting that the relationship we observe would be stronger with complete data. The results of these analyses are plotted in *Figure 1—figure supplement 5*.

## Modeling the determinants of epistasis

Because our modeling approach can be strongly influenced by outliers, we only consider fitness effect measurements for mutations with at least five cBCs across the two clones for these analyses (or three cBCs for the corresponding analysis in which clones are treated separately). First, we perform least-squares regression between background fitness and fitness effect for each mutation in each environment. As described in the main text, we classify mutations as being negatively or positively correlated with fitness based on both a statistical test (p<0.05, Wald test) and the effect size (|slope|>0.05). This heuristic slope cutoff is meant to filter for cases in which the range of fitness effects under consideration is larger than the typical noise for a single fitness effect measurement. In the YPD 30°C environment, a slope of 0.05 across an ~0.15 range of fitnesses will mean fitness effects should vary ~0.007, which is also the mean standard error of our fitness effect measurements in that environment (an analogous calculation in SC 37°C would yield a lower threshold, but we keep 0.05 for consistency).

Next, we fit our data with the three linear models described in the main text. To fix the intercepts of our models correctly, we first transform the fitness effect of each mutation by subtracting the mean fitness effect measured across all populations at the first timepoint of the evolution experiment (we denote each transformed fitness effect for mutation $m$ in population $p$, timepoint $t$, measured in condition $e$ as $\widetilde{s}_{m,p,t,e}$ below). We similarly transform our background fitness variable by subtracting the average fitness measurement across all populations at the first timepoint (we denote each transformed background fitness in population $p$, timepoint $t$, measured in environment $e$ as $\widetilde{x}_{p,t,e}$ below). Then we fix the intercept at (0, 0) in the modeling described below. Note that our plots showing these model fits use the natural scales for fitness and fitness effects, not these transformed scales. The fitness model (XM) can be written as

$$\widetilde{s}_{m,p,t,e} = \beta_{m,e}\widetilde{x}_{p,t,e} + e_{m,e}$$

where $e_{m,e}$ is a normally distributed noise term, and $\beta_{m,e}$ is a fitted parameter representing the slope between background fitness and the fitness effect of the mutation.

The idiosyncratic model (IM) can then be written as

$$\widetilde{s}_{m,p,t,e} = \sum_{p,t}\alpha_{m,p,t,e}i_{p,t,e} + e_{m,e}$$

$i_{p,t,e}$ is an indicator variable that is 1 at timepoints >= $t$ in population $p$ and zero in all other cases, and $\alpha_{m,p,t,e}$ is a fitted parameter associated with that indicator variable.

The full model (FM) can then be written as

$$\widetilde{s}_{m,p,t,e} = \beta_{m,e}\widetilde{x}_{p,t,e} + \sum_{p,t}\alpha_{m,p,t,e}i_{p,t,e} + e_{m,e}$$

In both idiosyncratic model and full model, the idiosyncratic epistasis terms ($\alpha_{m,p,t,e}i_{p,t,e}$) are added iteratively. At each step, we add the parameter that decreases the BIC the most if that decrease is more than 2. We do not allow parameters that fit only a single point, and we do not allow more than one parameter per population (i.e., the maximum number of idiosyncratic parameters for a given mutation in a given environment is the number of populations, six). Importantly, the idiosyncratic terms in the idiosyncratic model and the full model for a given mutation in a given environment may be different, so IM is not nested within FM and IM may explain more variance than FM in some cases. Because we consider the fixed intercept term to be part of our models, we compute $R^2$ for each model as 1 – (the sum of squared residuals/the *centered* sum of squares). This $R^2$ value can be negative if the model explains less variance than a model with only a free intercept term, in which case we set $R^2$ to zero and do not plot the model in *Figure 3* or *Figure 4*. All model fitting was performed by OLS using the Python package statsmodels (*Seabold and Perktold, 2010*).

To test how much noise affects our model fitting procedure, we ran our analysis on a simulated dataset and a shuffled dataset. In both cases, we focus on the sets of up to 36 fitness effect measurements associated with a mutation and a condition (with measurements in six populations and six timepoints). For the simulated dataset, we drew fitness effects for each set of mutations from a normal distribution with a mean of zero and a standard deviation equal to the mean empirical standard error for the fitness effects within the set of mutations (i.e., we drew all $s_{m,p,t,e}$ values for a given $m$ and $e$

from a normal distribution with a standard deviation of $mean\ (\sigma_{m,p,t,e})$). For the shuffled dataset, we randomly shuffled within each set (i.e., we randomly shuffled all $s_{m,p,t,e}$ values for a given $m$ and $e$).

The distributions of the coefficients obtained from our modeling procedure for the empirical, shuffled, and simulated datasets are plotted in *Figure 3—figure supplement 4*. In YPD 30°C, we found 291, 91, and 87 IM coefficients in our empirical data, shuffled data, and simulated data, respectively. In SC 37°C, we found 245, 78, and 81 IM coefficients in our empirical data, shuffled data, and simulated data, respectively. In clones isolated from evolution in YPD 30°C and assayed in SC 37°C, we found 241, 60, and 60 IM coefficients in our empirical data, shuffled data, and simulated data, respectively. In YPD 30°C, we found 140, 45, and 36 FM coefficients in our empirical data, shuffled data, and simulated data, respectively. In SC 37°C, we found 88, 38, and 46 FM coefficients in our empirical data, shuffled data, and simulated data, respectively. In clones isolated from evolution in YPD 30°C and assayed in SC 37°C, we found 199, 63, and 57 FM coefficients in our empirical data, shuffled data, and simulated data, respectively.

## Measuring background fitness

We measured the background fitness of clones with fluorescence-based competitive fitness assays in duplicate for each clone in each environment using the reference strains strain 2490A-GFP1 and 11470A-GFP1 for the YPD 30°C and SC 37°C clones, respectively. We used the 2490A-GFP1 reference when we assayed the YPD-30°C-evolved clones in SC 37°C because some of these clones have very low fitness and 2490A-GFP1 has a lower fitness than 11470A-GFP1. We used the fitness difference measured between these two references in *Johnson et al., 2021* to standardize the fitness measurements YPD-30°C-evolved clones in SC 37°C so that all fitness measurements in SC 37°C are on the same scale. Fitness assays were performed and data was analyzed as described in *Johnson et al., 2021*. Briefly, we maintained mixed cultures of our clones and fluorescent references for three daily growth cycles, as described above, and measured the frequency of fluorescent cells at each transfer using flow cytometry. We then calculated the fitness of each clone as the slope of the natural log of the ratio between the frequencies of the nonreference and reference cell populations over time. Finally, we calculated the mean and standard error of the fitness measurements for the two clones associated with each population-timepoint.

## Acknowledgements

We thank Sergey Kryazhimskiy, Alena Martsul, Andrew Murray, and members of the Desai lab for useful discussions about experimental design and analysis. We thank Shreyas Gopalakrishnan, Juhee Goyal, and Megan E Dillingham for their help with isolating the clones used in this experiment. We thank Craig Miller and one anonymous reviewer for helpful discussion and comments during the revision process. This work was supported by an NSF Graduate Research Fellowship (to MSJ), the NSF (PHY-1914916), and the NIH (GM104239). Computational work was performed on the Cannon cluster supported by the Research Computing Group at Harvard University.

## Additional information

### Funding

| Funder | Grant reference number | Author |
| --- | --- | --- |
| National Science Foundation | Graduate Research Fellowship | Milo S Johnson |
| National Science Foundation | PHY-1914916 | Michael M Desai |
| National Institutes of Health | GM104239 | Michael M Desai |

The funders had no role in study design, data collection and interpretation, or the decision to submit the work for publication.

## Author contributions

Milo S Johnson, Conceptualization, Formal analysis, Investigation, Writing – original draft, Writing – review and editing; Michael M Desai, Conceptualization, Resources, Supervision, Funding acquisition, Investigation, Writing – original draft, Project administration, Writing – review and editing

## Author ORCIDs

Milo S Johnson http://orcid.org/0000-0003-0169-2494
Michael M Desai http://orcid.org/0000-0002-9581-1150

## Decision letter and Author response

Decision letter https://doi.org/10.7554/eLife.76491.sa1
Author response https://doi.org/10.7554/eLife.76491.sa2

---

# Additional files

## Supplementary files

• Supplementary file 1. Column-annotated underlying data for this project. Includes background fitness, fitness effect, and modeling data from this experiment and *Johnson et al., 2019*.

• Supplementary file 2. Oligos used in this study.

• Transparent reporting form

## Data availability

Raw sequencing data has been deposited in the GenBank SRA (accession: SRP351176). All code used in this project is available on GitHub (https://github.com/mjohnson11/VTn_pipeline, copy archived at swh:1:rev:02d2b41d54dd22487df1c75f9e381411c5ef0376). All figures are based on data included in Supplementary File 1.

The following dataset was generated:

| Author(s) | Year | Dataset title | Dataset URL | Database and Identifier |
|---|---|---|---|---|
| Johnson MS, Desai MM | 2021 | Mutational robustness changes during long-term adaptation in laboratory budding yeast populations | https://www.ncbi.nlm.nih.gov/bioproject/PRJNA789529 | NCBI BioProject, PRJNA789529 |

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
