## [Editor Report]

Johnson and Desai developed an innovative yeast experimental-evolution system where they can insert barcoded disruptive mutations into the genome and measure their individual effect on fitness. They use this system to test whether these mutations have different effects on evolving lineages as they adapt over time. As expected, the mean fitness effect does decline in most (but not all) populations as lineages adapt, but in another condition, mean fitness effects of mutations do not change as the populations adapt. The authors suggest an intriguing interpretation that the ‘control coefficient’ of selection on growth can shift between different genetic modules over time, resulting in differing magnitudes of epistasis.

---

## [Decision Letter]

**Decision letter after peer review:**

Thank you for submitting your article "Mutational robustness changes during long-term adaptation in laboratory budding yeast populations" for consideration by *eLife*. Your article has been reviewed by 2 peer reviewers, and the evaluation has been overseen by a Reviewing Editor and Senior Editor. The following individual involved in the review of your submission has agreed to reveal their identity: Craig Miller (Reviewer #2).

Essential revisions:

(1) Justify the selection of the 91 mutants and how they are broadly representative. If they are not for the following reasons, considerable additional experiments are required.

In the authors' 2019 study, of ~1,147 insertion mutants assayed, ~91 of them had fitness effects distinguishable from neutral, i.e. ~8% are "effective". If (a) the percentage of "effective mutations" are similar between the genetic backgrounds used in the 2019 study and the current study, and (b) the genetic backgrounds used in these two studies are not correlated, then "effective mutations" from the 2019 study could still be "effective" only by chance. If the same percentage holds, then only 8% of 91 may (7 mutants) be drivers of the DFE or epistasis forms reported here. Though the paper suggests that 20-30 mutants are deleterious in the current dataset (figure 1 supplement 5), the lack of proper error estimates as mentioned below makes it difficult to evaluate these estimates

(2) Clarify the statistical power of various comparisons, as follows. Fitness is obtained by averaging effects on two clones from a population and variation between these measurements due to genetic differences or error are insufficiently accounted for. Some measures have r-squares <0.5, which begs the question, how much of the noise is coming from mutational differences between clones and how much from measurement noise? Was there any replication of fitness estimation on identical clones so that one has a good estimate of the measurement noise? How is noise accounted for and how does it influence the epistasis modeling? Much is made about the amount of the distribution with positive vs negative coefficients-but sampling noise alone will make coefficients deviate from zero.

3) Both reviewers and the editor were confused by the rationale and choice of the nested statistical framework, as noted in greater detail below by Reviewer 2. For example, It is unclear why for many mutations the idiosyncratic model explains more of the variance than the full model (e.g. Figure 3A). (Note, the fitness-mediated model never fits better than the full model). Also, when dealing with nested models in general, one should ask whether the more complex model fits the data enough better to justify accepting it over simpler model(s). There are clearly details and constraints in the models used here (and likely in the fitting process) that matter, but these are not discussed in any detail, or is the choice of model selection explained adequately.

*Reviewer #1 (Recommendations for the authors):*

Comments on figures:

Figure 1B: What is the error bar on y-axis? Without an error bar, conclusions cannot be made, particularly due to small fitness changes on y-axis (0.015 to 0.02 fitness changes while the difference between two clones from the same population is up to ~0.02). As acknowledged in Line 124, noise in fitness measurements contribute significantly to the modest differences in the DFE between clones.

Figure 2: Make it clear that each dot in the figure represents a disruption mutation, whose fitness is averaged on the background of two clones from the same timepoint/population. Define "background fitness". The label of x-axis is currently missing. Fitness errors on x-axis are not considered when inferring the significance of the correlation. Furthermore, the examples in Figure 2 are sufficient to conclude that epistasis is not strictly fitness-mediated. Authors could foreshadow the conclusions of the next section.

Figure 3: Line 214-215, walk the reader through the paper. Current writing is difficult to understand. What do the coefficients represent?

3B: Include a positive or negative model of IM. Consider also show models with x-axis as fitness to keep it consistent with 3C. Similarly, show examples with coefficient ~0 in 3C. The current selection of examples in 3B and 3C can lead to misunderstanding of the IM and Full-M.

One thing I don't quite understand is that models with more parameters as in IM or full-M are going to give better explanatory power than those with fewer parameters as in Fitness-M. Are there alternative ways to evaluate these models taking the number of parameters in the model into consideration?

Line-by-Line comments:

Line 81: Introduce the concept of mutational robustness. The immediate relation between mutational robustness and directional evolution experiments where bottleneck/drift likely plays little role in is not obvious. The authors should explain it in the introduction to motivate the rest of the manuscript.

Line 106: Explain why these 91 insertion mutations were chosen.

Line 108-112: Fitness correlation between clones from the same population is not enough to justify the averaging of their fitness as detailed in Public Review.

Line 118: How was p-value calculated? Was error of fitness measurements considered?

Line 159-160: What is the rationale underlying the choice of 0.05 for slope cutoff?

Line 203-210: Briefly describe the models, for instance, what (how many) parameters were used. If possible, include the formula of the model.

Line 235-236: Explain why the remaining epistatic effects are more likely to be negative.

Line 237-238: between the epistasis and environment ("GXGXE" effects).

Line 243: Which figure to look at?

Additional analysis that authors could consider:

Will the conclusion change if more/fewer time points from the evolving populations has been studied (for instance, clones were isolated from longer or shorter evolution)?

How could the magnitude of fitness increase during evolution affect the conclusion (i.e. the maximum fitness increase of clones from YPD is ~0.06 while the maximum fitness increase in SC is ~0.03)?

*Reviewer #2 (Recommendations for the authors):*

I am enthusiastic about this work. My main criticism is that the modeling is not laid out with enough care and thoroughness, leaving me to question some of the downstream conclusions. I have also offered many specific comments to the authors that are all in the form of comments within the manuscript pdf. I have emailed this annotated pdf to the editors who will upload it on my behalf.

[Editors' note: further revisions were suggested prior to acceptance, as described below.]

Thank you for resubmitting your work entitled "Mutational robustness changes during long-term adaptation in laboratory budding yeast populations" for further consideration by *eLife*. Your revised article has been evaluated by Aleksandra Walczak (Senior Editor) and a Reviewing Editor.

The manuscript has been improved but there are some remaining issues that need to be addressed, as outlined below:

Reviewer 1 identified issues with some of the supplemental figures, which must be fixed. However, we all appreciate this extensive revision and look forward to a final version.

*Reviewer #1 (Recommendations for the authors):*

The authors have sufficiently addressed the previous concerns over the choice of mutants and statistical analysis.

However, multiple supplement figures in the revised manuscript do not have any data points. The technical errors should be fixed before the publication.

I believe once the errors are fixed, the manuscript is ready to be published.

---

## [Author Response]

Essential revisions:(1) Justify the selection of the 91 mutants and how they are broadly representative. If they are not for the following reasons, considerable additional experiments are required.In the authors' 2019 study, of ~1,147 insertion mutants assayed, ~91 of them had fitness effects distinguishable from neutral, i.e. ~8% are "effective". If (a) the percentage of "effective mutations" are similar between the genetic backgrounds used in the 2019 study and the current study, and (b) the genetic backgrounds used in these two studies are not correlated, then "effective mutations" from the 2019 study could still be "effective" only by chance. If the same percentage holds, then only 8% of 91 may (7 mutants) be drivers of the DFE or epistasis forms reported here. Though the paper suggests that 20-30 mutants are deleterious in the current dataset (figure 1 supplement 5), the lack of proper error estimates as mentioned below makes it difficult to evaluate these estimates

We agree that if the identity of “effective mutations” was completely uncorrelated between the 2019 study and this one, this would imply that only ~8% of the 91 mutations (7 mutants) would have fitness effects in our data set. However, this would be surprising if true, since the strains used in the two studies are in fact quite similar. To be specific, the structure of the relatedness of the strains differs (reflecting laboratory adaptation in this study as compared to offspring of a cross in the 2019 paper), but the strains we consider in this paper are about as closely related to the segregants considered in the 2019 paper as those segregants are to each other.

To test this, prior to beginning this experiment we conducted a pilot study to confirm that these mutations had effects in the strains for this experiment (and our final dataset is in effect a more complete and less noisy version of this pilot). To explicitly show this comparison, we have added a new Figure Supplement (Figure 1 —figure supplement 8), which we reproduce below. As is clear from this figure, the range of fitness effects of the 91 mutations measured in this experiment and their effects in our 2019 experiment are quite similar (i.e. they are similarly “effective” in both studies). In fact, the percentage of fitness effect measurements that are significantly different from zero (after a Benjamini-Hochberg correction at the 0.05 level) is 57% here, compared to 48% in the previous experiment (we do not expect higher percentages since we often observe patterns of epistasis that include fitness effects near or at neutrality).

To make this point more clear we have also added an additional clarification in the main text: “This set of mutations was identified previously as a subset of random insertion mutations that have measurable effects in the strains from Johnson et al. (2019). These mutations have a similar spectrum of effects in the clones isolated for this study (Figure 1 —figure supplement 8), suggesting that they are also a broad sample of insertion mutations with measurable fitness effects in these strains.”

We also appreciate the comment on Figure 1 —figure supplement 5. The y-axis label “# of deleterious muts. measured” in our original draft was misleading: it is really a measure of the number of “strongly deleterious mutations” (defined as having a mean fitness effect < -0.05 across all population-timepoints in a given condition) that were successfully measured in each population-timepoint. We have added this phrasing to the main text, the figure itself, and the figure caption to clarify that this only shows strongly deleterious mutations (and relates only to the issue of missing measurements).

Finally, we note that any experiment of this type faces a fundamental limitation on the number of fitness effects that can be measured, and it is very likely that we would discover many more interesting cases of epistasis were we able to test the effects of a larger set of mutations over the course of these evolutionary trajectories. However, based on the analysis shown in Figure 1 —figure supplement 8, we believe these 91 mutations are comparably “effective” in these strains as they were in the previous experiment.

2) Clarify the statistical power of various comparisons, as follows. Fitness is obtained by averaging effects on two clones from a population and variation between these measurements due to genetic differences or error are insufficiently accounted for. Some measures have r-squares <0.5, which begs the question, how much of the noise is coming from mutational differences between clones and how much from measurement noise? Was there any replication of fitness estimation on identical clones so that one has a good estimate of the measurement noise? How is noise accounted for and how does it influence the epistasis modeling? Much is made about the amount of the distribution with positive vs negative coefficients-but sampling noise alone will make coefficients deviate from zero.

These are good questions, and we have taken several steps to clarify these statistical comparisons. First, we have more clearly described how we calculate the standard errors of *s* for mutations in each assay using the internal replication structure provided by the multiple redundant barcodes associated with each mutation (in Materials and methods; see below for revised text). These errors are now shown as error-bars in Figure 1 —figure supplements 1-3. Second, we describe how we calculate Benjamini-Hochberg corrected *p*-values describing the probability each mutation in each population-timepoint has a fitness effect of zero, which we include in our final dataset and use to address point 1 above. Both of these steps closely follow what was performed and described in Johnson et al. (2019).

Third, we have conducted a parallel analysis in which we treat each clone independently, as suggested by reviewer #1, and show that our conclusions are qualitatively unchanged. We have also changed the language in the main text to reflect the point you make here:

“Because the molecular dynamics of evolution in these haploid populations are characterized by successive selective sweeps, we expect the two clones isolated from each population at each timepoint to have very similar genetic backgrounds. When we compare the average fitness effect measurement for each insertion mutation between these clones, we generally see strong agreement, with a few exceptions (Figure 1 —figure supplements 1-3). These exceptions likely represent rare but important genetic differences between clones from the same population-timepoint. Given this, we chose to analyze our data in two ways. First, we improve the reliability of our fitness effects for each population-timepoint by using measurements from cBCs from both clones, treating them as we treated biological replicates in Johnson et al., 2019. Second, we treat each clone independently. Figure 1 —figure supplement 7, Figure 2 —figure supplement 5, Figure 3 —figure supplement 5, and Figure 4 —figure supplement 1 show that our qualitative conclusions are unchanged when using this second approach.”

Fourth, we have more carefully examined the possibility that noise in our measurements could affect the results of our modeling procedures. In the main text, we have included a more complete explanation of the procedures we use to prevent overfitting during modeling:

“Because mutations generally fix between every pair of timepoints during evolution, there could in principle be one idiosyncratic parameter for each timepoint in each population, but allowing all of these parameters would constitute over-fitting. To combat this possibility, we do not allow parameters that fit a single point (e.g., a parameter for an effect at the final timepoint), and we only allow one parameter per population.”

Even with these restrictions, we agree that noise in our fitness effect measurements is likely to give rise to some of the fitted parameters in our models. In order to assess the extent of this potential issue, we created both a simulated and a randomly-shuffled dataset and re-ran our modeling procedure on them. We explain how we constructed these datasets and how many coefficients resulted for each model in the Material and Methods. The distributions of these coefficients are plotted in Figure 3 —figure supplement 4. We find that while a considerable number of coefficients arise when we apply our models to these simulated and shuffled datasets, a larger number and a wider range of coefficients arise from our empirical data. This suggests that we should treat individual idiosyncratic coefficients with skepticism when looking at our data, but that we can still draw meaning from the overall patterns in these distributions. Given these results, we have added the following line to the main text: “While these coefficients can arise in our modeling procedure due to noise alone, we find far more coefficients in our empirical dataset than in a simulated or shuffled dataset (see Materials and methods, Figure 3 —figure supplement 4).” We have also removed our speculative discussion of the distribution of full model coefficients in SC 37°C, which is about a deviation in the distribution of coefficients that is similar to those we see in our simulated/shuffled dataset modeling. None of our conclusions rest upon this point; it was originally included as an interesting observation.

Here is the relevant new text in Materials and methods:

“To test how much noise affects our model fitting procedure, we ran our analysis on a simulated dataset and a shuffled dataset. In both cases, we focus on the sets of up to 36 fitness effect measurements associated with a mutation and a condition (with measurements in 6 populations and 6 timepoints). For the simulated dataset, we drew fitness effects for each set of mutations from a normal distribution with a mean of zero and a standard deviation equal to the mean empirical standard error for the fitness effects within the set of mutations (i.e. we drew all s_m,p,t,e_ values for a given m and e from a normal distribution with a standard deviation of mean(σ_m,p,t,e_)). For the shuffled dataset, we randomly shuffled within each set (i.e. we randomly shuffled all s_m,p,t,e_ values for a given m and e).

The distributions of the coefficients obtained from our modeling procedure for the empirical, shuffled, and simulated datasets are plotted in Figure 3 —figure supplement 4. In YPD 30°C, we found 291, 91, and 87 IM coefficients in our empirical data, shuffled data, and simulated data, respectively. In SC 37°C, we found 245, 78, and 81 IM coefficients in our empirical data, shuffled data, and simulated data, respectively. In clones isolated from evolution in YPD 30°C and assayed in SC 37°C, we found 241, 60, and 60 IM coefficients in our empirical data, shuffled data, and simulated data, respectively. In YPD 30°C, we found 140, 45, and 36 FM coefficients in our empirical data, shuffled data, and simulated data, respectively. In SC 37°C, we found 88, 38, and 46 FM coefficients in our empirical data, shuffled data, and simulated data, respectively. In clones isolated from evolution in YPD 30°C and assayed in SC 37°C, we found 199, 63, and 57 FM coefficients in our empirical data, shuffled data, and simulated data, respectively.”

We have added more clear descriptions of the modeling details to both the methods and Results sections. The counterintuitive phenomenon where the idiosyncratic model sometimes explains more variance than the full model is indeed due to the fitting process. Since idiosyncratic parameters are added iteratively and must reduce the Bayesian Information Criteria by at least 2 at each step, in some cases the full model (in which fitness is included as a parameter) has fewer parameters and/or less explanatory power than the idiosyncratic model. Another way to explain this is that while the fitness model is nested in the full model (both have fitness as a parameter), the idiosyncratic model is not nested in the full model (there may be parameters in the idiosyncratic model that are not in the full model).

3) Both reviewers and the editor were confused by the rationale and choice of the nested statistical framework, as noted in greater detail below by Reviewer 2. For example, It is unclear why for many mutations the idiosyncratic model explains more of the variance than the full model (e.g. Figure 3A). (Note, the fitness-mediated model never fits better than the full model). Also, when dealing with nested models in general, one should ask whether the more complex model fits the data enough better to justify accepting it over simpler model(s). There are clearly details and constraints in the models used here (and likely in the fitting process) that matter, but these are not discussed in any detail, or is the choice of model selection explained adequately.Reviewer #1 (Recommendations for the authors):Comments on figures:Figure 1B: What is the error bar on y-axis? Without an error bar, conclusions cannot be made, particularly due to small fitness changes on y-axis (0.015 to 0.02 fitness changes while the difference between two clones from the same population is up to ~0.02). As acknowledged in Line 124, noise in fitness measurements contribute significantly to the modest differences in the DFE between clones.

For Figure 1B, we have calculated standard errors on the DFE mean using the errors from the measurements of fitness effects that make up the distribution, and added this text to the methods:

“We calculate the standard error of the mean of the distribution of fitness effects by combining variances from two sources: error in our measurement of mutation fitness effects and errors in our measurement of mean fitness. Since this second component of variance is shared across mutations, it is not scaled by the number of mutations:

σDFEmean,p,t=∑mutationsσm,p,t,cbc2n2+σneut,p,t2 Where n is the number of mutations with measurements in the distribution (we use an analogous formula in our analysis on individual clones).”

We note that these errors do not consider missing measurements, which are likely to be important. These are discussed in the main text and explored more fully in Figure 1 —figure supplement 5. We have added text to the methods clarifying this analysis, and this edited section in the main text describes the results:

“As in Johnson et al., 2019, missing measurements of strongly deleterious mutations are more common in more-fit strains (clones from later timepoints) in YPD 30°C, suggesting that the negative correlation between generations evolved and the mean of the DFE would be stronger with complete data. Indeed, if we look at a limited set of mutations with measurements in every population-timepoint, or if we “fill in” missing measurements using their average fitness effect across population-timepoints, we see similar or stronger patterns of change (P<5×10^-7^, Wald Test) in the DFE mean in YPD 30°C (Figure 1 —figure supplement 5).”

Figure 2: Make it clear that each dot in the figure represents a disruption mutation, whose fitness is averaged on the background of two clones from the same timepoint/population. Define "background fitness". The label of x-axis is currently missing. Fitness errors on x-axis are not considered when inferring the significance of the correlation. Furthermore, the examples in Figure 2 are sufficient to conclude that epistasis is not strictly fitness-mediated. Authors could foreshadow the conclusions of the next section.

We now include error bars for standard errors on both axes of Figure 2, and we have changed the caption of Figure 2 to clearly describe the graphs:

“Each point depicts the fitness effect (y-axis) of one insertion mutation measured in one population-timepoint, with the measured fitness of that population-timepoint represented on the x-axis.”

We do not label the axes of the individual graphs in this figure to avoid clutter, instead labeling the mini graphs surrounding the figure. We think that along with the caption text above, the axes labels are well described.

You are correct that we do not consider the errors on the x-axis when doing least-squares regression on these data. During this revision we researched and implemented methods for errors-in-variables regression models such as orthogonal regression and the method described in York et al. (2004). While these methods can be used to more accurately infer the parameters of a linear relationship, they are not necessary for detecting *the presence* of a relationship, and we found that they can be sensitive to errors in our measurement of standard error (i.e. they can overfit points with especially low standard errors, which may sometimes arise by chance in our error calculations (which are now described in more depth in Materials and methods)). We ultimately determined that in this case, ordinary least squares (OLS) is an appropriate method, because (1) we have uncorrelated errors on the x and y axes, and (2) we are concerned with the presence of a relationship between variables rather than an accurate measurement of the slope of a known relationship. When using OLS, errors on either axis will tend to attenuate our estimates of the slope (towards zero), so this is a conservative approach.

We agree that this figure makes the point that epistasis is not strictly fitness-mediated. We have added this line to the start of the Modeling section: “The examples in Figure 2 demonstrate that correlations between fitness effects and fitness are common but often don’t explain the bulk of epistasis.”

Figure 3: Line 214-215, walk the reader through the paper. Current writing is difficult to understand. What do the coefficients represent?

We have rewritten this sentence to provide more clarity: “The examples in Figure 3B also demonstrate that epistasis is not strictly fitness-mediated; we commonly observe a stepwise change in the fitness effect of an insertion mutation in one evolving population, likely indicating epistasis between the insertion mutation and one or more mutations that fix in that population at a particular timepoint.”

The following sentence on the next page clarifies what the coefficients represent: “Positive and negative coefficients in the idiosyncratic model represent positive and negative epistasis between mutations that fix during evolution and our insertion mutations.”

3B: Include a positive or negative model of IM. Consider also show models with x-axis as fitness to keep it consistent with 3C. Similarly, show examples with coefficient ~0 in 3C. The current selection of examples in 3B and 3C can lead to misunderstanding of the IM and Full-M.

We agree that the figure should show more diverse examples of the modeling results. We have updated Figure 3 to include 4 examples of each model, showing a range of magnitudes of both positive and negative coefficients. While we appreciate the suggestion to use fitness on the x-axis, we have opted to continue using generations for the idiosyncratic model plots in order to emphasize that we can think of the determinants of fitness effects without referencing fitness, instead relying on how genotypes are related based on shared ancestry.

One thing I don't quite understand is that models with more parameters as in IM or full-M are going to give better explanatory power than those with fewer parameters as in Fitness-M. Are there alternative ways to evaluate these models taking the number of parameters in the model into consideration?

We have addressed this point above in the essential revisions section.

Line-by-Line comments:Line 81: Introduce the concept of mutational robustness. The immediate relation between mutational robustness and directional evolution experiments where bottleneck/drift likely plays little role in is not obvious. The authors should explain it in the introduction to motivate the rest of the manuscript.

We have added the following sentences to the beginning of the paragraph to introduce the concept of mutational robustness, with the relevant citations: “Robustness can be broadly defined as invariance in the face of perturbation (Masel and Siegel, 2009). Here we are concerned specifically with mutational robustness, a measure of how invariant phenotypes are to mutations (Lauring et al. 2013).”

Line 106: Explain why these 91 insertion mutations were chosen.

We now address this point as described above.

Line 108-112: Fitness correlation between clones from the same population is not enough to justify the averaging of their fitness as detailed in Public Review.

See our response to this point above.

Line 118: How was p-value calculated? Was error of fitness measurements considered?

We have updated the text here and elsewhere to indicate that Wald Tests were used to calculate *p*-values for least-squares regression. We discuss our choice to use ordinary least-squares (assuming equal errors) above.

Line 159-160: What is the rationale underlying the choice of 0.05 for slope cutoff?

This is a good question, we have added the following text to the main text and Methods to explain why we chose to use this heuristic cutoff:

In the main text: “The cutoff of 0.05 for the slope was chosen to filter for mutations with an effect size across the range of fitness that is larger than the typical standard error for our fitness effect measurements (see Materials and methods).”

In Materials and methods: “As described in the main text, we classify mutations as being negatively or positively correlated with fitness based on both a statistical test (P<0.05, Wald Test) and the effect size (|slope|>0.05). This heuristic slope cutoff is meant to filter for cases in which the range of fitness effects under consideration is larger than the typical noise for a single fitness effect measurement. In the YPD 30°C environment, a slope of 0.05 across an ~0.15 range of fitnesses will mean fitness effects should vary ~0.007, which is also the mean standard error of our fitness effect measurements in that environment (an analogous calculation in SC 37°C would yield a lower threshold, but we keep 0.05 for consistency).”

Line 203-210: Briefly describe the models, for instance, what (how many) parameters were used. If possible, include the formula of the model.

We have included a more in-depth description of each model, including formulas, in the Materials and methods.

Line 235-236: Explain why the remaining epistatic effects are more likely to be negative.

The point we were making here was that in a model including a background fitness effect (the full model), there are more negative than positive coefficients in P3 SC 37°C. However, since this was based on a relatively small number of coefficients, we have removed this sentence in the new draft (see response above about how noise can affect the fitted coefficients).

Line 237-238: between the epistasis and environment ("GXGXE" effects).

We have made this change, with slightly different wording: “between epistasis and the environment”.

Line 243: Which figure to look at?

We have included a reference to Figure 4A-B here.

Additional analysis that authors could consider:Will the conclusion change if more/fewer time points from the evolving populations has been studied (for instance, clones were isolated from longer or shorter evolution)?How could the magnitude of fitness increase during evolution affect the conclusion (i.e. the maximum fitness increase of clones from YPD is ~0.06 while the maximum fitness increase in SC is ~0.03)?

The relationships between the mean of the DFE and generations or background fitness remain significant (P<0.05, Wald Test, least-squares regression) in YPD 30°C when we exclude the first and second timepoints, but not once we also exclude the third timepoint. We believe this analysis is practically an analysis of power, which will decrease as the number of timepoints or the range in fitness increase changed, and we have included the data necessary for other researchers to ask this type of question in Supplementary File 1.

Reviewer #2 (Recommendations for the authors):I am enthusiastic about this work. My main criticism is that the modeling is not laid out with enough care and thoroughness, leaving me to question some of the downstream conclusions. I have also offered many specific comments to the authors that are all in the form of comments within the manuscript pdf. I have emailed this annotated pdf to the editors who will upload it on my behalf.

Thank you for the detailed comments here and in the attached pdf, and in particular for your deep engagement with the ideas in the Discussion and Ideas and Speculation sections. We have made numerous small edits throughout the paper based on your suggestions, and we have attached a pdf in which we respond to each of your comments along with this revision.

[Editors' note: further revisions were suggested prior to acceptance, as described below.]

The manuscript has been improved but there are some remaining issues that need to be addressed, as outlined below:Reviewer 1 identified issues with some of the supplemental figures, which must be fixed. However, we all appreciate this extensive revision and look forward to a final version.Reviewer #1 (Recommendations for the authors):The authors have sufficiently addressed the previous concerns over the choice of mutants and statistical analysis.However, multiple supplement figures in the revised manuscript do not have any data points. The technical errors should be fixed before the publication.

We believe the reviewer is referring to several blank plots in Figure 1 —figure supplements 1-3 and Figure 1 —figure supplement 9, which are due to missing data. It is also possible that there were PDF conversion issues that removed points from supplemental figures, which will be fixed for the final figure submission.

To clarify the blank plots, we have added the following to the figure captions for Figure 1 —figure supplements 1-3: “Blank graphs represent cases in which one or both replicates did not have fitness measurements, due to low read counts or insufficient neutral barcodes for mean fitness estimation.” We have also added the following to the figure caption for Figure 1 —figure supplement 9: “Assays in which less than three timepoints have at least 5000 reads are not plotted.”